# Phenolic compounds induce ferroptosis-like death by promoting hydroxyl radical generation in the Fenton reaction
Xinyue Sui[1], Jichao Wang[1], Zhiqiang Zhao[2], Bin Liu[3], Miaomiao Liu[3], Min Liu[1], Cong Shi[1], Xinjun Feng[2], Yingxin Fu[1], Dayong Shi[1], Shengying Li [1], Qingsheng Qi [1], Mo Xian[2] & Guang Zhao [1,2] ✉

Phenolic compounds are industrially versatile chemicals, also the most ubiquitous pollutants. Recently, biosynthesis and biodegradation of phenols has attracted increasing attention, while phenols' toxicity is a major issue. Here, we evolved phloroglucinol-tolerant *Escherichia coli* strains via adaptive evolution, and three mutations (Δ*sodB*, Δ*clpX* and *fetAB* overexpression) prove of great assistance in the tolerance improvement. We discover that phloroglucinol complexes with iron and promotes the generation of hydroxyl radicals in Fenton reaction, which leads to reducing power depletion, lipid peroxidation, and ferroptosis-like cell death of *E. coli*. Besides phloroglucinol, various phenols can trigger ferroptosis-like death in diverse organisms, from bacteria to mammalian cells. Furthermore, repressing this ferroptosis-like death improves phloroglucinol production and phenol degradation by corresponding strains respectively, showing great application potential in microbial degradation or production of desired phenolic compounds, and phloroglucinol-induced ferroptosis suppresses tumor growth in mice, indicating phloroglucinol as a promising drug for cancer treatment.

Phenolic compounds possess an aromatic ring bonded with one or more hydroxy groups, and represent a broad class of fine and bulk chemicals with diverse applications in the industrial and consumer fields. For example, phenol is the most economically important phenolic compound, widely used as a building block for the synthesis of various plastics, phenolic resins, synthetic fibers and bisphenol A[1], and its annual production is about 8.9 million tons worldwide[2]. Currently, phenol is derived from petrochemicals feedstocks such as benzene and toluene, which often raises sustainability, environmental, and economic issues[1]. As a more sustainable and feasible alternative, development of microbial cell factories to produce phenolic compounds from biomass-derived sugars has made great progress in past decades. Microbial production of phenol[3–5], catechol[6,7], pyrogallol[8,9], hydroquinone[10], and phloroglucinol[11,12] has been reported.

On the other side, phenolic compounds are among the main pollutants most ubiquitously distributed in industrial effluents, and can be accumulated in living organism. To treat phenols-containing wastewater, ozonation, hydrogen peroxide, UV and Fenton's reagent are commonly used, but these methods are environmentally harmful and inappropriate for large volumes of wastewater[13,14]. Instead of those, biodegradation is more promising due to its environmental friendliness and practical feasibility[13,14]. Up

to date, a number of microorganisms have proved to degrade phenols such as *Bacillus*[15,16], *Pseudomonas*[17,18], *Rhodococcus*[19], *Cupriavidus necator*[20], and *Escherichia coli* was engineered to remove phenol in wastewater through introduction of heterologous phenol degrading genes recently[21].

Unfortunately, most phenolic compounds are toxic to microorganism even at a low concentration. Growth inhibition of phenolic compounds (either end-products[3,4,8,22] or substrates[23,24]) has become a major limiting issue for commercialization of phenols-related biochemical processes. According to previous reports, phenolic compounds are known to damage the cell membrane for the hydrophobicity[25,26]. Besides, they could increase the generation of reactive oxygen species (ROS), leading to denaturation of proteins/enzymes and damage of DNA and cytoskeleton[27]. Some anti-oxidant enzymes, such as glutathione-disulfide reductase, superoxide dis-mutase (SOD), laccase, and alkyl hydroperoxide reductase, were induced by the presences of phenolic compounds and believed to contribute to resis-tance of phenols-induced oxidative stress in various species[23,24,28,29]. It is baffling how phenols increase the intracellular ROS level as they are highly reductive chemicals themselves, and understanding the detailed mechanism of phenols-induced ROS generation will help to develop phenols-tolerant strain.

[1]State Key Laboratory of Microbial Technology and Institute of Microbial Technology, Shandong University, Qingdao, China. [2]CAS Key Lab of Biobased Materials, Qingdao Institute of Bioenergy and Bioprocess Technology, Chinese Academy of Sciences, Qingdao, China. [3]TEDA Institute of Biological Sciences and Bio-technology, Nankai University, Tianjin, China. ✉e-mail: zhaoguang@sdu.edu.cn

*E. coli* is one of the most studied bacteria, and was engineered to produce the phenolic compound phloroglucinol (PG) in our group previously[11]. In this study, *E. coli* and PG were adapted as proxy to elucidate the molecular mechanism of phenols' toxicity. Three genes/operon, *clpX*, *sodB* and *fetAB*, were identified essential to *E. coli* PG tolerance, and it is discovered that PG can complex with iron and promote generation of hydroxyl radical (HO·) in the Fenton reaction, inducing ferroptosis-like cell death of *E. coli*. This ferroptosis-like death pathway can be triggered by various phenolic compounds in diverse organisms, from bacteria to mammalian cells. Furthermore, we demonstrate that repressing the phenols-induced ferroptosis-like death benefits bacterial degradation or production of desired phenolic compound, and PG-induced ferroptosis can suppress tumor growth in mice, showing great application potential in both biotechnology and healthcare fields.

## Results

### Identification of *E. coli* genes essential to PG tolerance

*E. coli* BL21(DE3) was grown in minimal salt medium (MSM) to late-exponential phase ($OD_{600}$ 2.5, Supplementary Fig. 1), and then treated with 0.5, 1.0, 2.0, and 4.0 g/L PG, respectively. About 5% cells survived after challenge of 0.5 g/L PG, and all cells were killed with presence of 4 g/L PG (Fig. 1a), showing that PG is highly toxic to *E. coli*. Adaptive laboratory evolution is a powerful method to elevate bacteria tolerance against environmental stresses and to identify genes essential to stress tolerance[30,31]. In this study, adaptive evolution was carried out using the GREACE method[32], in which mutated *dnaQ* gene (encoding proofreading factor of the DNA polymerase) was introduced into BL21(DE3) strain for in vivo continuous mutagenesis. After 52 rounds of selection, PG concentration increased from 1 g/L to 10 g/L (Fig. 1b), reaching its solubility, and the survival culture was spread onto LB agar plate to isolate single colonies. Then six colonies with the highest PG tolerance, which also showed improved phenol tolerance,

were designated M01-06 and subjected to genome sequencing along with the parent strain Q3505.

The sequences of both evolved mutants and the parent strain Q3505 were aligned to the *E. coli* BL21(DE3) reference sequence (NC_012978.2) to figure out the genetic mutations arose during evolution process. Totally, 28 mutations were identified in these six strains including 13 single-nucleotide polymorphisms (SNPs) and 15 insertions/deletions (Supplementary Table 1). Among these mutations, SNP in *zipA* gene resulted in a synonymous variant, gene *B* encodes a λ phage capsid protein, and *lacZ*, *rbsD* and *insB27* are nonfunctional pseudogenes, so mutations related with these five genes were omitted from further characterization. To identify the contribution of each mutation to the improved PG tolerance, knockout mutants of genes *fhuA*, *clpX*, *flgK*, *sodB*, *arcZ* and *yigB*, site-directed mutants *apaH* P237Q and *ftsZ* R214H, and SNP mutants in the intergenic regions of *qmcA-fetA*, *nfuA-gntT* and *yshB-glnG* were constructed in the parent strain Q3505, respectively. Following PG tolerance assay, three mutations (Δ*clpX*, Δ*sodB* and *qmcA-fetA* SNP) proved of great assistance in the tolerance improvement (Fig. 1c).

As shown in Fig. 1d, *qmcA* gene and *fetAB* operon are divergent and the SNP locates in their shared promoter region. qRT-PCR results demonstrated that this SNP enhanced the mRNA level of *fetAB* about 20 times while it has no effect on *qmcA* transcription (Fig. 1e). Furthermore, *qmcA* and *fetAB* were cloned into vector pTrcHis2B and introduced into wild-type strain, respectively. The strain carrying p*qmcA* presented a survival rate similar with strain carrying empty vector, whereas overexpression of *fetAB* amazingly restored the viability of *E. coli* BL21(DE3) strain in treatment with PG (Fig. 1f).

The transcription start site of *fetA* in SNP mutant was mapped using RACE experiment. Different with the start site S1 reported previously[33], *fetA* has a new transcription initiation site S2 located at 27 bp upstream of the start codon, and the SNP site is exactly situated in the putative -35 box of S2

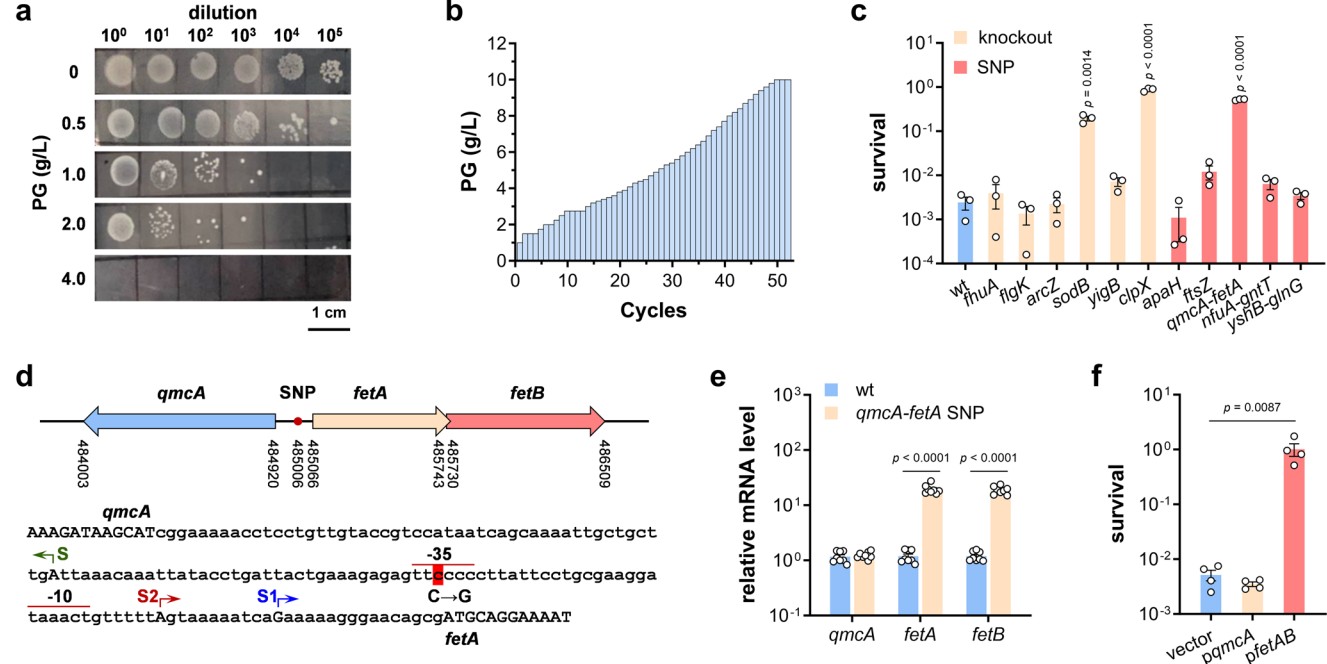

**Fig. 1 | Identification of *E. coli* genes related with PG tolerance using adaptive laboratory evolution. a** Tolerance of *E. coli* BL21(DE3) strain after treatment with PG at different concentrations for 4 h. A representative result was shown. **b** PG concentrations used in each round of adaptive evolution process. **c** Tolerance of *E. coli* BL21(DE3) wild-type strain and defined knockout and SNP mutants after treatment with PG at 1.3 g/L for 4 h (*n* = 3 biological independent samples). **d** Schematic diagram showing *qmcA* and *fetAB* genomic locations and sequence of their shared promoter region. The SNP mutation, previously reported *fetA*

transcription starting site S1, newly identified initiation site S2 and its corresponding -35 and -10 regions were shown. **e** Relative mRNA level of *qmcA*, *fetA* and *fetB* determined by qRT-PCR in BL21(DE3) wild-type strain and *qmcA-fetA* SNP mutant (*n* = 4 biological independent samples with two technical repeats). **f** Tolerance of BL21(DE3) strains carrying empty vector, p*qmcA* and p*fetAB* after treatment with PG at 1.3 g/L for 4 h (*n* = 4 biological independent samples). Two-tailed Student's tests were performed to determine the statistical significance for two group comparisons. Error bars, mean ± standard error of mean (SEM).

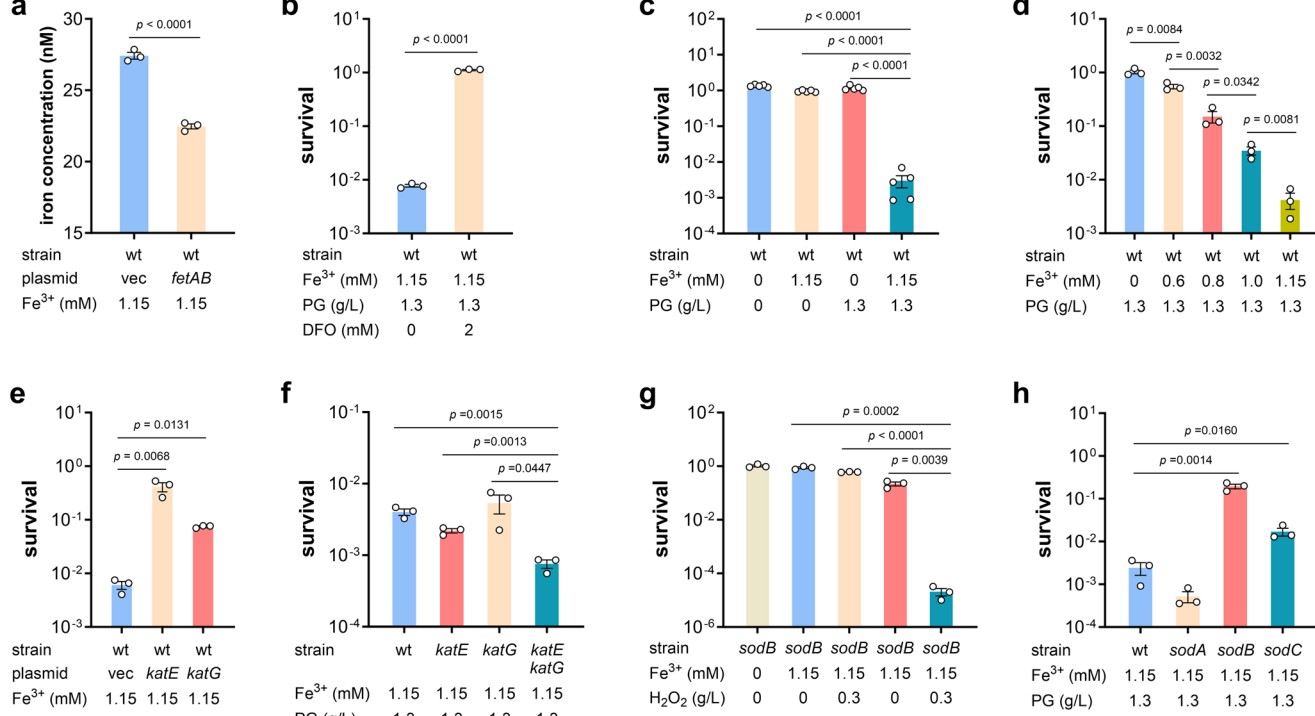

**Fig. 2 | Iron and $H_2O_2$ were required for PG toxicity. a** Intracellular iron concentration in BL21(DE3) strain carrying empty vector and p*fetAB* ($n = 3$ biological independent samples). **b** Survival rates of BL21(DE3) wild-type strain after treatment with PG at 1.3 g/L for 4 h with the absence and presence of the cell-permeable iron chelator desferrioxamine (DFO) ($n = 3$ biological independent samples). **c** Survival rates of BL21(DE3) wild-type strain in iron-free medium after challenge of PG, Fe, and both of them ($n = 5$ biological independent samples). **d** Survival rates of BL21(DE3) wild-type strain after exposure to 1.3 g/L PG and different concentrates of iron ($n = 3$ biological independent samples). **e** Survival rates of BL21(DE3) wild-type strains carrying empty vector, p*katE* and p*katG* after treatment with PG at 1.3 g/L for 4 h ($n = 3$ biological independent samples). **f** Survival rates of BL21(DE3) wild-type strain and *katE*, *katG*, and *katE katG* mutants after treatment with PG at 1.3 g/L for 4 h ($n = 3$ biological independent samples). **g** Survival rates of the *sodB* mutant after challenge of PG, $H_2O_2$, and both of them ($n = 3$ biological independent samples). **h** Survival rates of BL21(DE3) strain and knockout mutants of genes encoding SODs after treatment with PG ($n = 3$ biological independent samples). Two-tailed Student's tests were performed to determine the statistical significance for two group comparisons. Error bar, mean ± SEM.

promotor (summarized in Fig. 1d), which is the reason why this SNP mutation increased the transcription level of *fetAB* gene. Following, the function of *clpX*, *sodB*, and *fetAB* was confirmed by knockout and complementation (Supplementary Fig. 2). In summary, three genes/operon *clpX*, *sodB* and *fetAB* are essential to bacterial PG tolerance.

## Iron and hydrogen peroxide are required for PG toxicity

The *fetAB* operon encodes an iron exporter, and overexpression of *fetAB* reduced intracellular iron levels in *E. coli* (Fig. 2a). Taking into account that BL21(DE3) strain carrying p*fetAB* are super tolerant to PG (Fig. 1f), iron is supposed to facilitate the PG toxicity. To illustrate iron's role in PG killing, the cell-permeable iron chelator desferrioxamine was used to block intracellular iron. It turned out that desferrioxamine efficiently protected BL21(DE3) cells from treatment with PG (Fig. 2b). Subsequently, iron-free medium was used in PG killing assay, and iron and PG were supplemented into the medium alone or together. As shown in Fig. 2c, *E. coli* BL21(DE3) cells were killed only with the presence of both iron and PG. Furthermore, the cell viability declined in an iron concentration-dependent manner after exposure to 1.3 g/L PG and different concentration of iron (Fig. 2d). Hence, iron is essential for the PG toxicity, and *fetAB* overexpression enhanced exportation of intracellular iron, resulting in improved bacterial PG tolerance.

The *sodB* gene encodes a FeSOD, and *E. coli* also has two other SODs: manganese- and copper, zinc-cofactored SODs (MnSOD and CuZnSOD encoded by *sodA* and *sodC* genes, respectively)[34]. It is confusing that Δ*sodB* mutant presented higher PG tolerance as SOD genes were upregulated upon phenol challenge and considered as antioxidant enzymes to resist phenol-induced oxidative stress in previous studies[23,28]. Superoxide radicals form as

a harmful byproduct of aerobic metabolism, and is degraded by the SODs to hydrogen peroxide which is further converted into dioxygen by catalase KatG and KatE[34]. Based on the above information, the PG-tolerant *sodB* mutant should have a much lower intracellular concentration of hydrogen peroxide than wild-type strain, indicating that hydrogen peroxide is required for PG toxicity.

To test this hypothesis, *katE* and *katG* genes were cloned into plasmid vector pTrcHis2B and overexpressed in BL21(DE3) wild-type strain respectively, leading to a great enhancement of PG tolerance of this strain (Fig. 2e). In addition, *katE katG* double mutant was more susceptible to PG than wild-type strain (Fig. 2f). Furthermore, hydrogen peroxide was added into medium in *sodB* mutant's PG killing assay. While neither hydrogen peroxide nor PG killed cells alone, their combination decreased the survival rate of *sodB* mutant dramatically (Fig. 2g). Next, knockout mutants of the other two genes encoding SODs were constructed, and the survival rate of *sodC* mutant was found to be 10 times higher than that of wild-type strain while the *sodA* mutant did not affect the PG susceptibility significantly (Fig. 2h). All these results demonstrated that hydrogen peroxide is required for PG toxicity, and SODs impaired *E. coli* tolerance to PG as it produces hydrogen peroxide.

## PG-iron complex promotes the generation of hydroxyl radical

$H_2O_2$ can be reduced by iron via Fenton reaction to HO· that causes severe DNA, protein, and lipid damage[35]. In vivo experiments in *E. coli* suggested that DNA damage required $H_2O_2$ at a concentration of 1 to 2 mM[36], which could only be achieved in catalase deficient *E. coli* strain[35]. Furthermore, the majority of iron inside the cells forms complex with iron-storage proteins FtnA, Bfr, and Dps that limit the potential for iron-dependent HO·

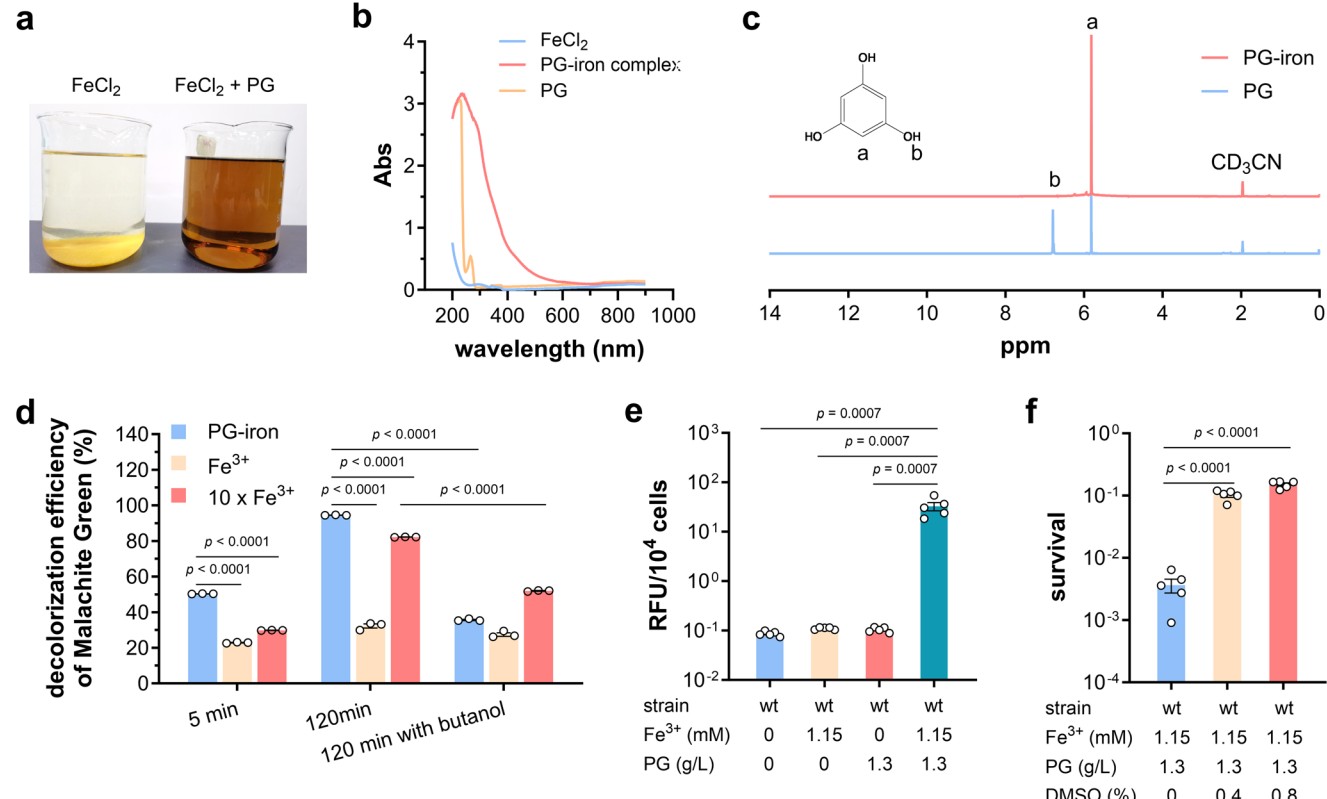

**Fig. 3 | PG-iron complex promoted the generation of hydroxy radicals (HO·) in Fenton reaction. a** Characteristics of ferrous chloride solutions at initial pH 7.0 with the absence and presence of PG. **b** UV-visible absorption spectra of solutions containing ferrous chloride alone, PG alone, and ferrous chloride and PG coupled at pH 7.0. **c** $^1$H NMR spectra of PG and PG-iron complex in acetonitrile-$d_3$ solution. **d** Decolorization of Malachite Green with Fenton reaction containing 36 µM PG-iron complex, 36 and 360 µM $FeCl_3$, respectively ($n = 3$). The decolorization efficiency after treatment of 5 min, 120 min, and 120 min with HO· scavenger $t$-butanol was shown. **e** Intracellular HO· concentration determined using hydroxyphenyl fluorescein of BL21(DE3) wild-type strain grown in iron-free medium after challenge of PG, Fe, and both of them ($n = 5$ biological independent samples). **f** Survival rates of BL21(DE3) wild-type strain after challenge of PG and Fe with dimethyl sulfoxide (DMSO) at different concentrations ($n = 4$ biological independent samples). Two-tailed Student's tests were performed to determine the statistical significance of two group comparisons. Error bar, mean ± SEM.

formation[35,37,38]. So, the Fenton reaction is not significant in normal *E. coli* cells.

As reported previously, the generation of HO· in Fenton reaction can be promoted by a series of complexes composed of iron and salen/salophene, in which complexation of iron takes place by phenolic groups[39,40]. Therefore, it was wondered whether PG could complex with iron and stimulate the Fenton reaction. In $FeCl_2$ solution with a pH of 7.0, $Fe^{2+}$ could be oxidized to $Fe^{3+}$ by dissolved oxygen, forming light brown flocs $Fe(OH)_3$. With addition of PG gradually, the precipitate got dissolved to form homogeneous brown aqueous solution (Fig. 3a). This phenomenon was believed to be caused by the complexation between $Fe^{3+}$ and PG, which was confirmed by the following UV-visible absorption assay (Fig. 3b) and NMR analysis (Fig. 3c and Supplementary Fig. 3).

To test whether PG-iron complex can enhance the production of HO· in Fenton reaction, Malachite Green (MG) decolorization experiment was carried out, in which MG will be oxidized by HO· and turned into colorless. As shown in Fig. 3d, the MG decolorization rate reached to 50.3% after 5 min with the Fenton reaction system containing 36 µM PG-iron complex, 2.2- and 1.7-time higher than with normal Fenton systems containing 36 and 360 µM $FeCl_3$, respectively. After 120 min, 94.5% of MG were oxidized by the PG-iron complex-containing Fenton system, but more than two thirds of MG remained with the same concentration of $FeCl_3$. To confirm that MG is oxidized by HO· rather than other substances, the HO· scavenger $t$-butanol was added into the MG decolorization reaction, and severely inhibited the oxidation of MG (Fig. 3d). All these results demonstrated that formation of the PG-iron complex could promote the production of HO· in Fenton reaction dramatically.

To identify the effect of PG-iron complex in vivo, *E. coli* BL21(DE3) strain was cultured in iron-free minimal medium supplemented with iron and/or PG, and the intracellular levels of HO· were determined. The cells grown with both iron and PG presented a HO· concentration at least 230-time higher than the others (Fig. 3e). Coupled with the fact that the HO· scavenger dimethyl sulfoxide rescued BL21(DE3) cells from the iron/PG killing assay (Fig. 3f), all above results demonstrated that the generation of HO· promoted by PG-iron complex was the main factor of PG toxicity to *E. coli*.

## Knockout of *clpX* gene increases the protein stability of RpoS and Dps

Our results showed that knockout of *clpX* gene significantly enhanced *E. coli* tolerance to PG (Fig. 1c). ClpX is an ATP-dependent molecular chaperone in the ClpXP protease complex, in which the ClpP subunits form the proteolytic center and ClpX serves as a substrate-specifying adaptor to unfold and transfer the substrate protein to the catalytic site[41]. RpoS is the major regulator of general stress response in *E. coli*. In growing cells, RpoS has an extremely short half-life due to its proteolysis by ClpXP, and various stress conditions led to stabilization and accumulation of RpoS. In addition, a direct recognition factor RssB having specific affinity for RpoS and targeting RpoS to ClpXP is essential to the proteolysis of RpoS (Fig. 4a)[42,43].

To test whether the increased PG tolerance of *clpX* mutant is related with the RpoS proteolysis, *rpoS* and *rssB* knockout mutants were constructed and subjected to PG tolerance assay. Although *rpoS* deletion did not affect cell growth (Supplementary Fig. 4), the *rpoS* mutant became much more susceptible to PG. In contrast, the PG tolerance of *rssB* mutant was

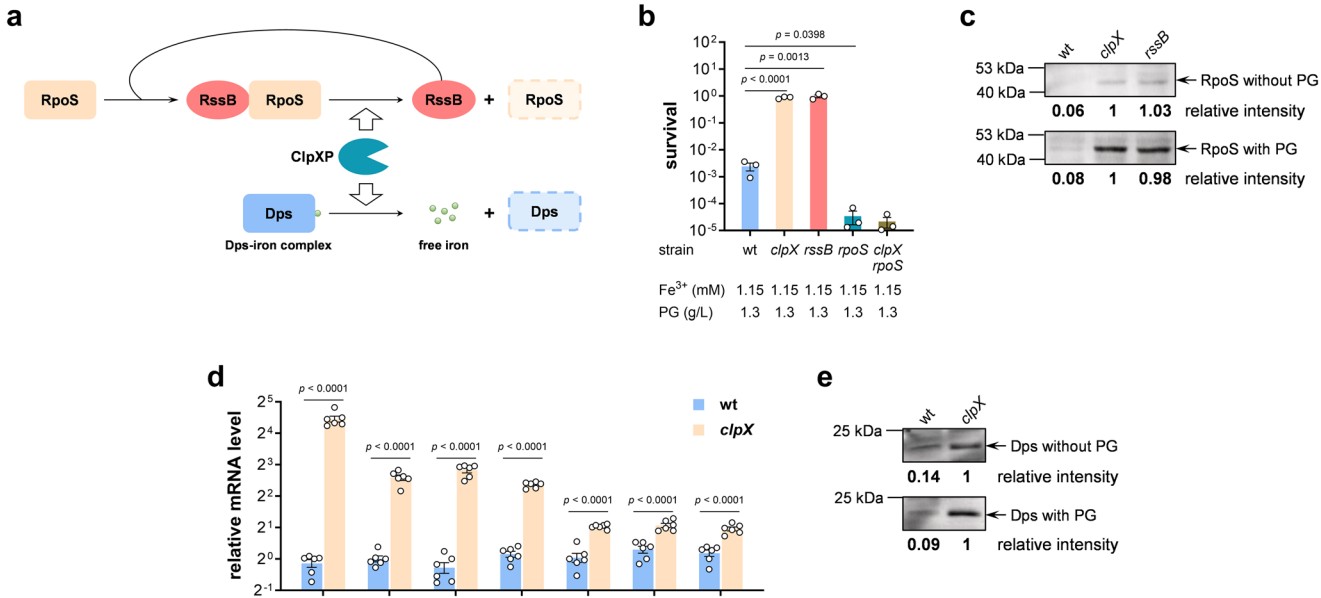

**Fig. 4 | Knockout of *clpX* gene improved the protein stability of general stress response regulator RpoS and miniferritin Dps. a** Model illustrating the proteolysis of RpoS and Dps proteins mediated by ClpXP protease. **b** Survival rates of *E. coli* BL21(DE3) wild-type strain, *clpX*, *rssB*, *rpoS*, and *clpX rpoS* mutant strains after treatment with PG and iron for 4 h ($n = 3$ biological independent samples). **c** Western blot of RpoS protein in BL21(DE3) wild-type, *clpX* and *rssB* mutants with the absence and presence of PG. **d** Relative mRNA level of RpoS-dependent oxidative stress response genes in BL21(DE3) wild-type and *clpX* strains ($n = 3$ biological independent samples with two technical repeats). **e** Western blot of Dps protein in BL21(DE3) wild-type and *clpX* mutant with the absence and presence of PG. Two-tailed Student's tests were performed to determine the statistical significance for two group comparisons. Error bar, mean ± SEM.

significantly higher, similar with that of the *clpX* mutant (Fig. 4b). Furthermore, a double mutant *clpX rpoS* strain presented similar susceptibility to PG as the *rpoS* mutant, suggesting that the function of *clpX* mutation to increase *E. coli* PG tolerance is dependent on the RpoS protein.

RpoS protein level was determined by Western blot using a rabbit polyclonal anti-RpoS antibody. In *E. coli* cells at mid-log phase no matter the presence of PG, knockout of both *clpX* and *rssB* gene enhanced the RpoS protein level remarkably (Fig. 4c and Supplementary Fig. 5). Furthermore, *clpX* deletion notably enhanced the transcription of several RpoS-dependent genes involved in bacteria oxidative stress response including *katE*, *uspB*, *yaiA*, *bsmA*, *yggE*, *yodB*, and *ychH* (Fig. 4d). Especially, the *katE* mRNA level was 21 times higher in the *clpX* mutant, which certainly would protect *E. coli* from PG toxicity (Fig. 2a).

Besides RpoS, the ClpXP protease can also degrade the Dps protein, which is a miniferritin and sequesters free iron[44]. To monitor the Dps level, a $his_6$-tag was fused to the C-terminal of Dps protein. As shown in Fig. 4e and Supplementary Fig. 6, much more Dps was detected in the wild-type strain than in the *clpX* mutant, that would help to reduce the intracellular concentration of free iron and the production of HO·. In summary, knockout of *clpX* stabilized the general stress response regulator RpoS and miniferritin Dps, and further rescued *E. coli* cells from oxidative stress.

### Phenolic compounds induce ferroptosis-like death in diverse organisms

Above results demonstrated that excessive HO· produced in Fenton reaction killed *E. coli* cells, similarly to ferroptosis, an oxidative and iron-dependent regulated cell death that was described in eukaryotic organisms[45,46]. To confirm that PG induces ferroptosis-like death of *E. coli*, several assays were carried out. Lipid peroxidation is one of hallmarks of ferroptosis, and leads to accumulation of malondialdehyde[46]. As shown in Fig. 5a, the presence of PG in iron-containing medium increased the malondialdehyde level of *E. coli* cells by more than 70 times. Furthermore, PG exposure of *E. coli* cells resulted in a dramatic decline of the reduced glutathione (GSH) (Fig. 5b), which is a characteristic of ferroptosis in animals[45]. In consistent, addition of GSH restored the viability of *E. coli* cells in treatment with PG (Fig. 5c).

Additionally, ferroptosis inhibitor ferrostatin-1 prevented *E. coli* cell death induced by PG (Fig. 5d). In animals, glutathione peroxidase 4 (GPX4) is a major ferroptosis defense system, detoxifying lipid hydroperoxides and inhibiting ferroptosis[47,48], and *E. coli* also encodes a GPX4 homolog protein BtuE, sharing an identity of 35.5% and a similarity of 62.3% with human GPX4 (Supplementary Fig. 7). So, the *btuE* gene was cloned into plasmid vector pTrcHis2B and overexpressed in BL21(DE3) strain. The strain carrying empty vector presented a survival rate similar with that of BL21(DE3) wild-type strain, while overexpression of *btuE* dramatically increased *E. coli* tolerance to challenge of PG and iron (Fig. 5e). All experiments suggested that PG induced the ferroptosis-like cell death of *E. coli*.

Next, some other phenolic compounds including phenol, catechol, resorcinol, pyrogallol, and naphthol, were tested on their abilities to induce *E. coli* ferroptosis-like death. Results in Fig. 5f showed that these phenolic chemicals could not kill *E. coli* BL21(DE3) strains alone, and supplementation of iron decreased the survival rates of BL21(DE3) strain one- to four-order of magnitude. Moreover, the toxicity of PG and phenol to bacteria *Salmonella typhimurium* and *Klebsiella pneumoniae* and eukaryote *Saccharomyces cerevisiae* was enhanced dramatically by the addition of iron as well (Fig. 5g). Although gram-positive bacteria are not tested, they are reported to be more susceptible to oxidative-ferroptotic death caused by iron complexes[39]. Most importantly, combination of PG and iron presented significant cytotoxicity to HeLa cells, while killing effect to HeLa cells was not detected with supplementation of PG or iron alone (Fig. 5h). All these results collectively demonstrated that phenolic compounds are capable to trigger the ferroptosis-like cell death pathway in diverse organisms, from bacteria to mammalian cells.

### Repressing phenols-induced ferroptosis-like death benefits both biodegradation and biosynthesis of phenolic compounds

Phenolic compounds are industrially versatile commodity chemicals, also the most ubiquitous pollutants[49,50]. Recently, biosynthesis of desired phenolic compounds and biodegradation of phenolic pollutants have attracted more attentions due to their environmental friendliness and practical feasibility. However, most phenolic compounds are toxic to microorganism,

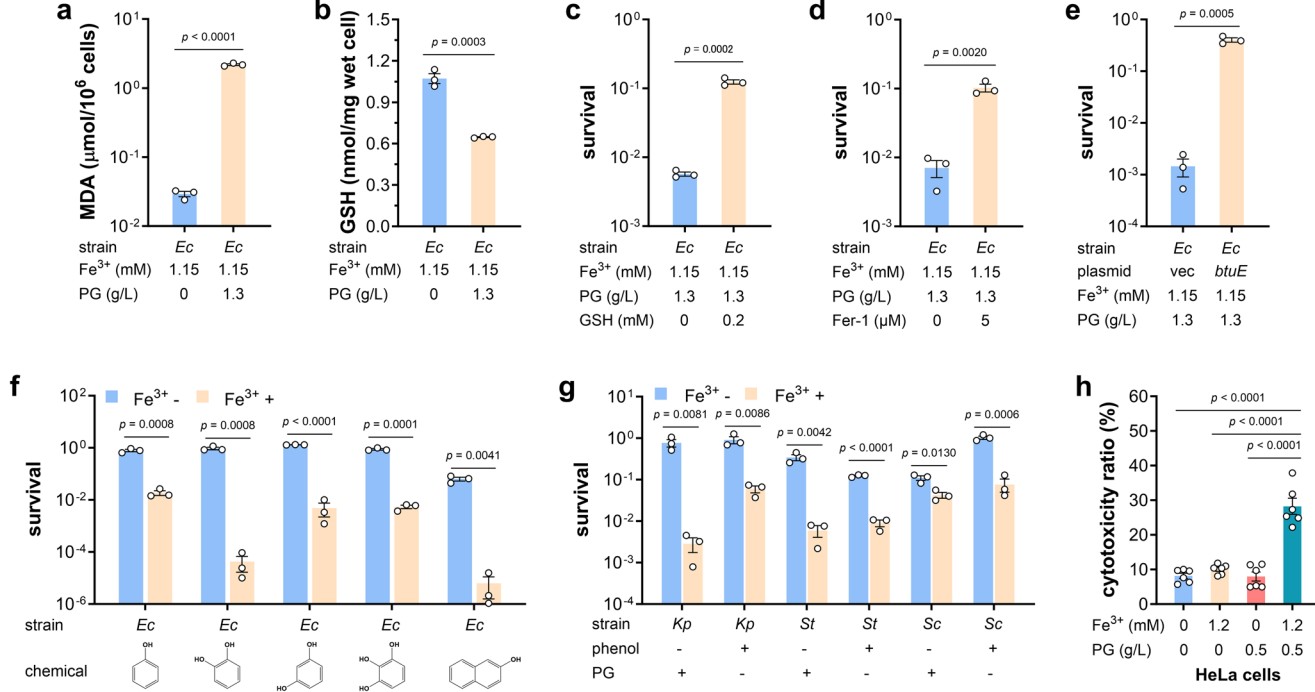

**Fig. 5 | Phenolic compounds induce ferroptotic cell death in diverse species.**
**a** Accumulation of malondialdehyde (MDA) in *E. coli* BL21(DE3) cells grown in iron-containing medium after treatment with PG ($n = 3$ biological independent samples). *Ec*, *E. coli*. **b** Intracellular level of reduced glutathione (GSH) in *E. coli* BL21(DE3) cells grown in iron-containing medium after treatment with PG ($n = 3$ biological independent samples). **c** Survival rates of *E. coli* BL21(DE3) cells after challenge of PG and iron with the absence and presence of GSH ($n = 3$ biological independent samples). **d** Survival rates of *E. coli* BL21(DE3) cells after challenge of PG and iron with the absence and presence of ferroptosis inhibitor ferrostatin-1 (Fer-1) ($n = 3$ biological independent samples). **e** Survival rates of *E. coli* BL21(DE3)

strain carrying empty plasmid vector and p*btuE* after treatment with PG and iron ($n = 3$ biological independent samples). **f** Survival rates of *E. coli* BL21(DE3) strain upon exposure to different phenolic compounds with the absence and presence of iron. Concentration of each compound was indicated. ($n = 3$ biological independent samples). **g** Survival rates of *Klebsiella pneumoniae* (*Kp*), *Salmonella typhimurium* (*St*), and *Saccharomyces cerevisiae* (*Sc*) upon exposure to PG and phenol with the absence and presence of iron ($n = 3$ biological independent samples). **h** The cytotoxicity of iron and PG to HeLa cells ($n = 6$ biological independent samples). Two-tailed Student's tests were performed to determine the statistical significance for two group comparisons. Error bar, mean ± SEM.

and growth inhibition of phenols has become a major limiting issue for commercialization of phenols-related biochemical processes[3,8,22,24].

To test whether this phenols-induced ferroptosis-like death affects phenol biodegradation process, *Pseudomonas* sp. DHS3Y strain was grown in M9 medium with 1 g/L phenol as sole carbon source under iron depleted- and iron rich-conditions, respectively. As shown in Fig. 6a, the strain under iron depleted-condition started to grow after 48 h, and reached an OD600 of 2.52 ± 0.04 after 72 h of inoculation, while its growth was not observed under iron rich conditions. In accordance with this, the phenol concentration decreased to 0.86 ± 0. 01 g/L after 60 h and phenol could barely be detected after 72 h of inoculation in iron depleted-medium, however there were still 0.93 ± 0. 01 g/L phenol left in the iron rich-medium after 72 h of inoculation (Fig. 6b). These results suggested that repression of phenol-induced ferroptosis-like death significantly improved the phenol biodegradation.

Furthermore, a PG-tolerant *E. coli* BL21(DE3) triple mutant strain (Δ*clpX*, Δ*sodB*, *qmcA-fetA* SNP) was constructed (Supplementary Fig. 8). Then the plasmid pA-*phlD/marA/acc* carrying PG biosynthetic pathway[12] was introduced into BL21(DE3) wild-type strain and triple mutant to generate strains Q3595 and Q4333, respectively. In shaking flask cultivation, the strain Q3595 produced 1.30 ± 0.03 g/L and 0.93 ± 0.02 g/L PG with the absence and presence of iron, whereas supplementation of iron in the strain Q4333 culture increased the PG production from 1.91 ± 0.03 g/L to 2.53 ± 0.05 g/L (Fig. 6c). Differently from this, iron enhanced the cell density of both strains in varying degrees (Fig. 6d). These results indicate that the toxicity of end-product PG is a major limiting factor for the wild-type strain Q3595, and the absence of iron impairs the ferroptosis-like death of *E. coli*, resulting in higher production of PG. However, for the resistant strain Q4333, PG is no longer a limiting factor and iron is essential for normal cell

growth and metabolism. For example, aconitase and fumarase in TCA cycle need iron as cofactor, and iron depletion remarkably decreased their enzymatic activities (Fig. 6e). These also proved the necessity of resistant strain in bioproduction of phenolic compounds, and it is not practicable to increase bacteria tolerance to phenols simply by growing in an iron-free environment.

## PG-induced ferroptosis suppresses tumor growth
Ferroptosis is considered as a new and promising option for cancer therapy as it remains functional in tumor cells that escape other forms of cell death such as apoptosis, necroptosis and pyroptosis[45,46,51]. To explore whether PG-induced ferroptosis can suppress tumor growth in vivo, tumor-bearing mice were generated by subcutaneous injection of human lung cancer H1299 cells and treated daily with water (control group), PG and PG-iron complex for 11 days as indicate in Fig. 7a. Administration of PG itself seemed to have some effect of suppressing tumor growth, but the statistically significant difference between control group and PG-treated group was not observed. Excitingly, complexation of PG and iron resulted in the lowest tumor growth rate and weight (Fig. 7b, c), suggesting that PG-iron complex dramatically reduced the H1299 tumor growth.

To confirm that ferroptosis was induced in tumor, tissue sections were stained with Prussian blue to detect the intracellular level of ferric iron, and results indicated the iron accumulation in tumors treated with PG-iron complex, but not in the other two groups treated with water and PG respectively (Fig. 7d). Lipid peroxidation is a hallmark of ferroptosis, producing some toxic products including 4-hydroxynonenal (4-HNE)[52], while GPX4 is an effective defense system protecting biomembranes from peroxidation damage[47,48]. So, GPX4 and 4-HNE are both considered as

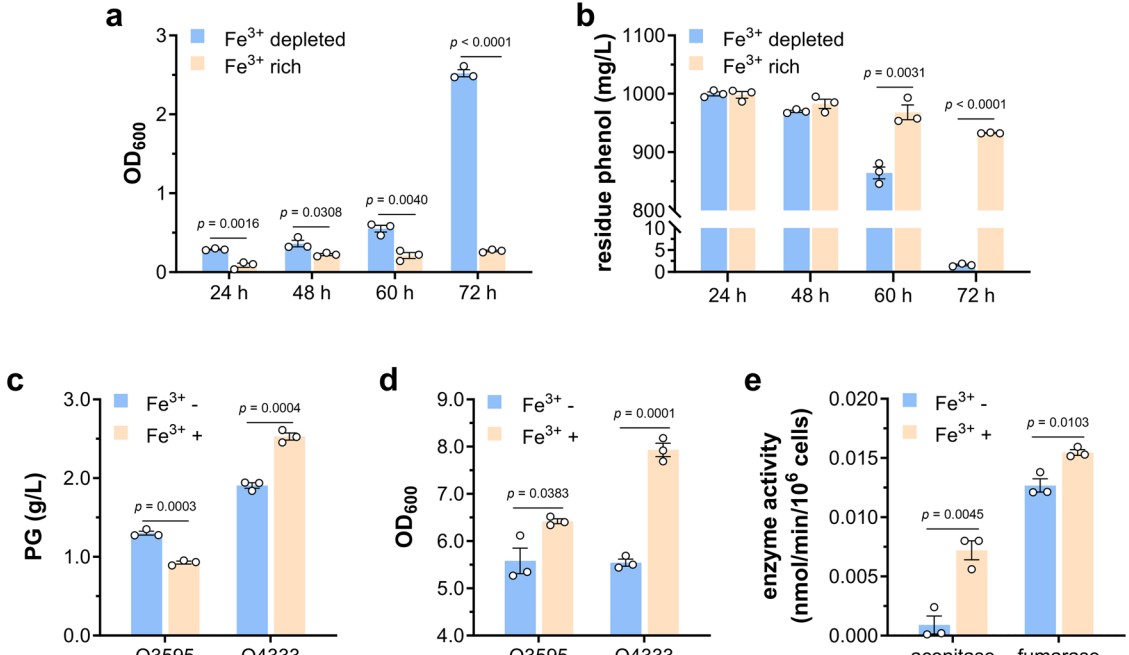

**Fig. 6 | Bacterial tolerance contributes to biodegradation and biosynthesis of phenols. a** Growth of *Pseudomonas* sp. DHS3Y in M9 medium using phenol as sole carbon source under iron depleted and iron rich conditions ($n = 3$ biological independent samples). **b** Degradation of phenol by *Pseudomonas* sp. DHS3Y under iron depleted and iron rich conditions ($n = 3$ biological independent samples). **c** PG production of Q3595 (*E. coli* BL21(DE3) carrying PG biosynthetic pathway) and Q4333 (PG tolerant mutant carrying PG biosynthetic pathway) grown in MSM with

the absence and presence of iron ($n = 3$ biological independent samples). **d** Growth of Q3595 and Q4333 in MSM with the absence and presence of iron ($n = 3$ biological independent samples). **e** Activity of aconitase and fumarase tested using whole cell lysate of *E. coli* BL21(DE3) cells grown with the absence and presence of iron ($n = 3$ biological independent samples). Two-tailed Student's tests were performed to determine the statistical significance for two group comparisons. Error bar, mean ± SEM.

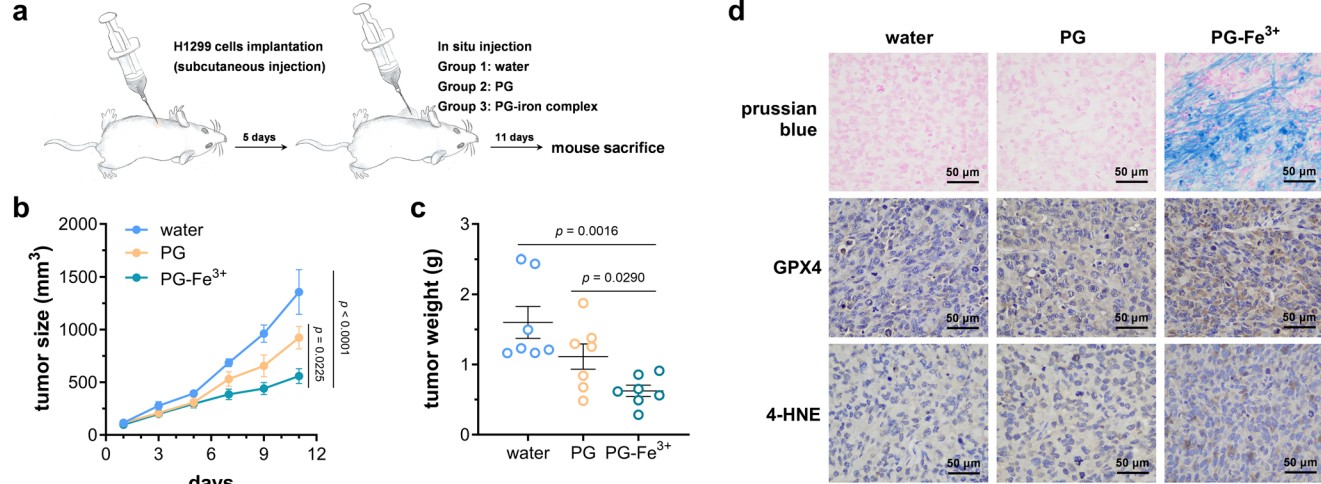

**Fig. 7 | PG-induced ferroptosis suppressed tumor growth. a** Schematic experimental procedure for implantation of BALB/C nude mice and treatment with PG and PG-iron complex. **b** Growth of H1299 tumors in mice treated with water (control group), PG and PG-iron complex. Seven biological independent samples were tested in this experiment, and two-way ANOVA assays were performed to determine the statistical significance. **c** Weights of H1299 tumors in mice treated with water, PG and PG-iron complex, measured on day 11 ($n = 7$ biological

independent samples). Two-tailed Student's tests were performed to determine the statistical significance. Error bars, mean ± SEM. **d** Representative staining and immunochemical images from H1299 tumors in mice treated with water, PG and PG-iron complex. Prussian blue staining was used to detect ferric iron. Glutathione peroxidase 4 (GPX4) and 4-hydroxynonenal (4-HNE) were detected using corresponding antibodies. Scale bar represents 50 μm.

biomarkers to monitor ferroptosis, and their level in tumors was determined by immunohistochemistry analysis. As shown in Fig. 7d and Supplementary Fig. 9, expression of GPX4 and accumulation of 4-HNE in tumors treated with PG-iron complex were much higher than those in control group and tumors treated with PG. These results proved that PG-iron complex-induced ferroptosis can suppress tumor growth effectively.

## Discussion

According to the results shown above, we discovered a previously uncharacterized mechanism by which phenolic compounds can form complex with iron ions and induce ferroptosis, a form of regulated cell death, in diverse species from bacteria to human cells (Fig. 8). Elevation of intracellular ROS level is one of the most prominent hallmarks of ferroptosis, and it

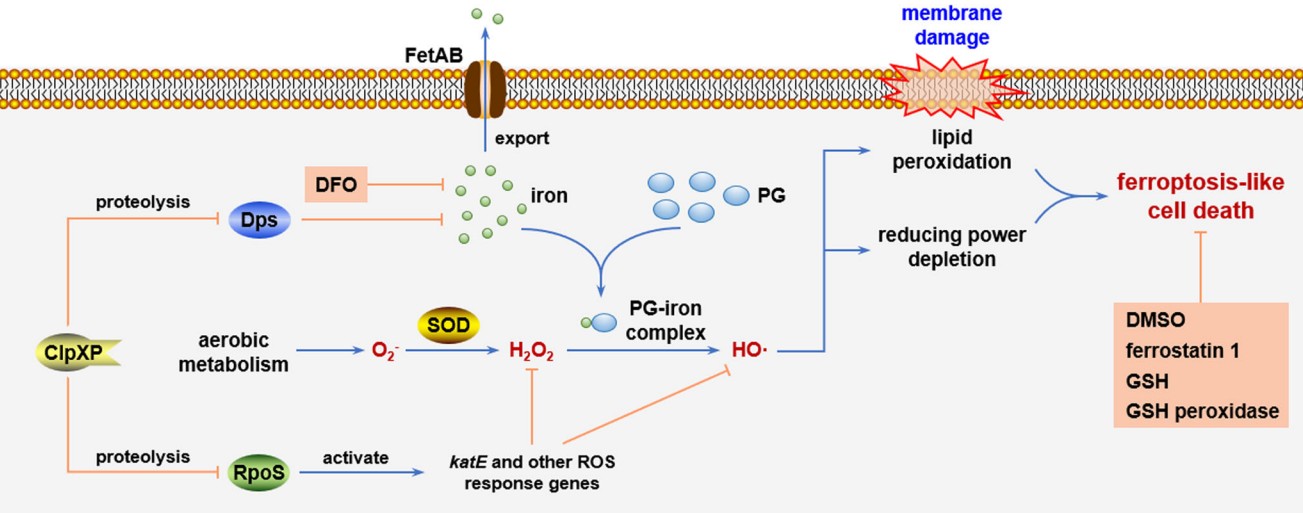

**Fig. 8 | Model illustrating the phenol-iron complex-induced ferroptosis-like death and functions of related genes.** Phloroglucinol (PG) can complex with iron and promote generation of hydroxyl radical (HO·) in the Fenton reaction, inducing ferroptosis-like cell death of *E. coli*. Iron exporter FetAB and superoxide dismutase (SOD) are involved in the intracellular supply of Fenton reaction substrates iron and hydroxy peroxide, respectively. Deletion of the protease ClpXP enhances *E. coli* tolerance to PG by increasing the stability of general stress response regulator RpoS and miniferritin Dps. This PG-induced ferroptosis-like death can be inhibited by iron chelator desferrioxamine (DFO), HO· scavenger DMSO, ferroptosis inhibitor ferrostatin 1, reduced glutathione (GSH) and overexpression of GSH peroxidase.

was proved in this study that phenolic compounds complex with iron and promote the generation of HO· in the Fenton reaction in vitro and in vivo (Fig. 3), leading to lipid peroxidation (Fig. 5a) and depletion of reducing power (Fig. 5b), and further leading to cell death. This cell death triggered by phenols and iron can be inhibited by supplementation of reducing equivalent (Fig. 5c), iron chelator desferrioxamine (Fig. 2b) and ferroptosis inhibitor ferrostatin-1 (Fig. 5d). These characteristics markers of eukaryotic ferroptosis were found to be conserved in *E. coli*, suggesting that ferroptosis may be an ancient cell death mechanism. We also reveal that three genes/ operon are involved in bacterial phenol tolerance: two of them, *sodB* and *fetAB*, are related with supply of substrates of Fenton reaction, hydrogen peroxide and iron respectively, and ClpXP protease determines the stability of RpoS protein, an essential regulator in bacterial oxidative stress response, and miniferritin Dps (Fig. 8). Our findings greatly enriched our under-standing of the molecular mechanism of phenols toxicity and the networks contributing to biological tolerance to phenolic compounds.

As reported, phenolic compounds can cause cell membrane damage[25,26] and ROS generation[27]. In this study, extremely high intracellular level of HO·, the most reactive and toxic one in ROS, was observed in *E. coli* cells challenged by PG and iron (Fig. 3d). Phenols are highly reductive chemicals themselves, and it is baffling how they cause the increase of intracellular ROS. Here we have provided sufficient experimental evidence to illustrate the mechanism of phenols-induced ROS production, in which phenols can form complex with iron and promote the Fenton reaction, resulting in accumulation of excessive HO· and ferroptosis-like cell death. Consistent with our results, the antioxidative responses have been dis-covered in various organisms exposed to phenols[23,24,28,29]. The phenolic compounds tested in this study are simple molecules with only one or two benzene rings, and they can induce ferroptosis-like death in diverse organisms. In contrast, polyphenols, natural products commonly found in higher plants, show antioxidant activity and inhibit ferroptosis[53]. This divergence is probably due to the molecular structure differences between simple phenolic compounds and polyphenols which usually contains sev-eral benzene rings and different functional groups.

Differently from previous reports, the cell membrane damage caused directly by phenols was unobvious in our study as lowering the HO· level by HO· scavenger DMSO (Fig. 3e) and inhibiting the PG-iron complex induced ferroptosis (Fig. 5c, d) could totally restore the viability of *E. coli* upon treatment with PG. Instead, we would prefer that the cell membrane

damage is also a consequence of the high intracellular HO· level, such as lipid peroxidation showed in the ferroptosis-like death process of *E. coli* (Fig. 5a) and eukaryotic cells[54].

We discovered that deletion of genes encoding SODs increased bac-terial tolerance to phenols (Fig. 2d), which is inconsistent with the previous knowledge that SODs, as members of antioxidant enzymes, contributed to resistance of phenols-induced oxidative stress in some microorganisms[23,24,28,29]. In *E. coli*, transcription of *sodA* gene is upregulated by SoxRS, principal regulator of the superoxide radical response[55], and the *sodB* gene is positively regulated by Fur through a small RNA RyhB[56]. So, it is very likely that the SOD genes are activated upon exposure to PG and iron. However, based on our results, we consider that SOD acts as an amplifier of phenols' toxicity in some senses: phenols promote the generation of HO· via Fenton reaction, inducing the expression of SODs; and SODs convert intracellular superoxide radicals into hydrogen peroxide, providing sub-strate for Fenton reaction. This positive feedback loop will aggravate the accumulation of intracellular HO·, eventually leading to cell death.

We proved the necessity of iron in the toxicity of phenolic compounds. Actually, the involvement of iron metabolism in bacterial response to phenol has been reported in several omic studies. For example, in *P. putida* KT2440 strain, the iron uptake repressor protein Fur was upregulated[28], and biosynthesis and uptake of the *Pseudomonas* characteristic siderophore pyoverdine were downregulated upon phenol exposure[57]. Moreover, *Rho-dococcus opacus* PD630 strain highly expressed the iron-storage protein Dps in response to phenol[58]. All these results indicated that bacteria lower the intracellular free iron levels in different ways upon phenol challenge, but the mechanism how iron metabolism affects bacterial phenol tolerance remains elusive. Our data provide strong evidence that formation of iron-phenols complex was the determining factor of Fenton reaction to generate HO· with a lethal concentration in intracellular niche.

Our results evidenced that antioxidant response governed by RpoS protein played an essential role in bacterial resistance to phenols. The RpoS ($\sigma^s$) subunit of RNA polymerase is the master regulator of the general stress response in *E. coli* and some other bacteria[42,43], and controls transcription of approximately 500 *E. coli* genes[59,60]. In exponentially growing *E. coli* cells, RpoS is expressed at a basal level and hardly detectable because of efficient proteolytic degradation dependent on ClpXP protease and recognition factor RssB[60]. Consistent with this, the RpoS level in *clpX* and *rssB* mutants was remarkably higher than in the wild-type *E. coli* strain (Fig. 4c), and then

the stabilized RpoS protein increased transcription of a series of oxidative stress response genes (Fig. 4d), including *katE* which is the most important part of the RpoS regulon in response to ROS[61] and whose overexpression significantly improved the *E. coli* tolerance to PG (Fig. 2a). Besides RpoS, OxyR and SoxRS also play a critical role in the defense for oxidative stress in *E. coli*[55], and they may be involved in the bacterial tolerance to phenols.

We demonstrate phenol-iron complex as an inducer of ferroptosis-like death in diverse organisms, like the previously reported salen/salophene iron complexes[39,40]. Our results have great application potential in different fields. Firstly, our results showed that *Pseudomonas* presented higher phenol tolerance and could degrade more phenol under iron depleted conditions, and a tolerant *E. coli* strain produced more PG that the wild-type strain (Fig. 6), suggesting that higher bacterial tolerance benefits these phenols-related biochemical processes. Phenolic compounds are industrially versatile commodity chemicals, also the most ubiquitous pollutants[49,50]. Recently, biodegradation and biosynthesis of phenolic compounds has attracted more and more attention due to the environmental friendliness and practical feasibility. However, growth inhibition of phenolic compounds (either end-products[3,4,8,22] or substrates[23,24]) has become a major limiting issue for commercialization of phenols-related biochemical processes, and alleviating the phenols' toxicity in microorganisms is an urgent task. Our results here will shed light on design and construction of phenols-tolerant bacteria strain, and further help the promotion and application of bioremediation of phenols polluted environment and biosynthesis of value-added phenolic compounds. In addition, bacteria with high tolerance to phenols can also be used in bioconversion processes of lignocellulosic biomass to biofuels and chemicals, as phenolic compounds are toxic by-products in the pretreatment and saccharification of lignocellulosic biomass, and repress the following fermentation[26].

Secondly, our results also have potential applications in the field of healthcare. Our data suggested that PG-induced ferroptosis is able to suppress tumor growth (Fig. 7). Cancer is a leading cause of death worldwide, and 609,820 cancer deaths are projected to occur in the United States in 2023[62]. To specifically kill cancer cells and remain normal cells alive, a principal approach is triggering regulated cell death of cancer cells[63,64]. Unfortunately, cancer cells can often escape several forms of regulated death such as apoptosis, necroptosis, and pyroptosis[45,46,51]. On the other hand, cancer cells maintain a higher intracellular iron level than non-cancer cells to enable rapid proliferation, making cancer cells more vulnerable to ferroptosis[63,64]. So, identification of chemicals, especially FDA-approved drugs, as ferroptosis inducers will pave the way of ferroptosis to be a promising treatment to kill therapy-resistant cancers. PG is already a widely used analgesic and smooth muscle antispasmodic[65], and our results suggested that PG is promising to become a drug for cancer treatment (Fig. 7). Based on known pharmacokinetics and safety profiles[66], repositioning PG for previously unknown indications can save costs and time for drug development, maximizing the use of existing resources.

## Methods
### Bacterial strains and growth conditions
All bacterial strains and plasmids used in this study are listed in Supplementary Table 2, and all primers used are synthesized by Sangon Biotech (Shanghai) Co. Ltd., and listed in Supplementary Data 2. Luria-Bertani broth (Oxoid) was used for the construction of strains and plasmids. For tolerance assay and PG production, strains were grown in shaking flask using MSM containing 20 g/L glucose, 9.8 g/L $K_2HPO_4 \cdot 3H_2O$, 3.0 g/L $(NH_4)_2SO_4$, 2.1 g/L citrate monohydrate, 0.3 g/L ferric ammonium citrate, 0.24 g/L $MgSO_4$, and 1 mL of trace element solution (3.7 g/L $(NH_4)_6Mo_7O_{24} \cdot 4H_2O$, 2.9 g/L $ZnSO_4 \cdot 7H_2O$, 24.7 g/L $H_3BO_3$, 2.5 g/L $CuSO_4 \cdot 5H_2O$, and 15.8 g/L $MnCl_2 \cdot 4H_2O$). In iron-free experiment, ferric ammonium citrate was omitted. When necessary, antibiotics were added at final concentrations of 100 μg/mL for ampicillin, 20 μg/mL for chloramphenicol, 100 μg/mL for spectinomycin, 100 μg/mL for apramycin or 50 μg/mL for kanamycin.

Plasmids are constructed by standard restriction cloning or using ClonExpress Ultra One Cloning Kit (Vazyme Biotech) based on in vitro multiple fragments recombination, and confirmed by DNA sequencing. Strains harboring chromosomal mutations were generated using pCas/pTargetF system[67] or pKSI-1/pREDTKI system[68].

For phenols challenge experiments, the strains were grown in MSM to an $OD_{600}$ of 2.5. When necessary, ferric ammonium citrate (0.3 g/L) or $FeCl_3$ (0.2 g/L), hydroxy peroxide (0.3 g/L), DMSO (0.4% and 0.8%), GSH (0.2 mM), ferrostatin-1 (5 μM) and desferrioxamine (2 mM) were added into the culture, and expression of plasmid carried genes was induced by 0.5 mM IPTG at an $OD_{600}$ of 0.8, and the strain was further grown to an $OD_{600}$ of 2.5. Phenolic compounds were added into the culture, and the cells were further grown for another 4 h. Specifically, PG concentration is 1.3 g/L for *E. coli* BL21(DE3), and for assay of *K. pneumoniae* and *S. typhimurium*, 8 g/L PG and 2.5 g/L phenol were used. For *S. cerevisiae*, 8 g/L PG and 4.2 g/L phenol were used. Then the cells were collected to determine the CFU, mRNA and protein levels. The intracellular iron and hydroxyl radical concentrations were determined using Iron Colorimetric Assay Kit (Applygen Technologies, China) and Hydroxyphenyl fluorescein (Shanghai Maokang Biotechnology, China), respectively. Lipid peroxidation is measured using Malondialdehyde Assay kit (Solarbio, Beijing, China). The GSH concentration was assayed using GSH and GSSG Assay kit (Beyotime, China).

The HeLa cell line (ATCC CCL-2) was grown in Dulbecco's modified Eagle's medium supplemented with 10% fetal bovine serum and penicillin-streptomycin at 37 °C in a 5% $CO_2$-containing atmosphere. The cells inoculated in 12-well plates ($3 \times 10^4$ cells/well) were cultured in normal medium for 16 h, followed by a culture in serum-free medium with 0.2 g/L $FeCl_3$ and/or 0.5 g/L PG for 4 h. Then cytotoxicity of Fe and PG was determined using CytoTox 96 Non-radioactive Cytotoxicity Assay kit (Promega).

### Adaptive evolution of *E. coli* BL21(DE3) strain
Adaptive evolution of BL21(DE3) strain was performed using the GREACE method[32] with modification. Briefly, a random mutant library of *dnaQ* gene (encoding proofreading element of the DNA polymerase complex) was introduced into BL21(DE3) strain for in vivo continuous mutagenesis. The strain was grown in MSM to an OD600 of 2.5, challenged with 1 g/L PG for 8 h, and recovered in LB broth overnight. Then LB culture was used as the inoculum for the next round of selection with higher PG concentration. To isolate evolved mutants, culture was spread onto LB agar plate, and single colonies were subjected to PG tolerance assay. Six colonies with the highest PG tolerance were designated as M01-06, and their genome were resequenced by BGI.

### Quantitative RT-PCR and RACE
Total RNA was isolated from bacterial culture using EASYSpin Plus Bacterial RNA kit (Aidlab Biotechnologies, China) according to the manufacturer's guide. Genomic DNA was removed and cDNA was synthesized using Evo M-MLV RT Kit with gDNA Clean for qPCR II (Accurate Biology, China). qRT-PCR was carried out using SYBR Green Pro Taq HS qPRC Kit (Accurate Biology, China) with the QuantStudio 1 system (Applied Biosystems). To normalize total RNA quantity of different samples, constitutively transcribed gene *rpoD* was used as a reference control. The relative amount of mRNA level was calculated using the ΔΔCt method. Three independent biological samples with two technical repeats for each sample were performed. The RACE experiment was carried out using SMARTer RACE 5′/3′ Kit (Clontech) according to the manufacturer's instructions.

### Immunoblot analysis
The *rpoS* gene was cloned into vector pET28a and expressed in *E. coli* BL21(DE3) strain, and RpoS protein was purified by Ni-affinity chromatography and used to immune rabbit to produce polyclonal anti-RpoS antibody. The *E. coli* cells were disrupted by sonication and centrifuged, and the supernatants were used for Western blot. Protein concentration in cell lysate was determined using BCA protein assay kit (Pierce), and the same amount of protein was loaded onto 12% SDS-PAGE, transferred to PVDF

membrane (Millipore), and incubated with anti-RpoS antibody and HRP-conjugated goat-anti-rabbit antibody (Abbkine, China). Then protein signal was detected using Immobilon Western HRP substrate (Millipore) and Fusion FX6 Imaging System (Vilber, France). To detect Dps-his6 protein, HRP Anti-6X His tag antibody [GT359] (Abcam, catalog No. ab184607, 1:10,000) was used. The relative amount of each band in Western blot is determined using ImageJ (NIH).

## Preparation and analysis of PG-iron complex decolorization of malachite green

$FeCl_2$ solution (pH 7) was allowed to stand aerobically until precipitation occurs, and PG solution was supplemented gradually until the precipitate dissolves. PG and PG-iron complex structures were confirmed by NMR analysis using an Avance Neo 600 NMR spectrometer (Bruker, Switzerland). The $FeCl_2$ solution, PG solution and PG-iron complex solution were scanned using U-2900 UV-visible Spectrophotometer (Hitachi, Japan). MG decolorization took place at room temperature in 50 mL of reaction system containing 0.1 g/L MG, 1 mM $H_2O_2$, 36 µM $FeCl_3$ or 360 µM $FeCl_3$ or 36 µM PG-iron complex. The reaction was quenched with addition of 50 mM t-butanol, and concentration of residue MG was determined at 618 nm using spectrophotometer.

## In vivo tumor growth and treatment

Human lung cancer H1299 cell line was obtained from National Collection of Authenticated Cell Cultures, Shanghai, China. H1299 cells were grown in RPMI 1640 medium supplemented with 10% fetal bovine serum. Male BALB/C nude mice (5 weeks old, Ziyuan Experimental Animal Co., Ltd, Hangzhou, China) were injected subcutaneously with $5 \times 10^6$ H1299 cells in PBS, and 21 mice with tumor size of 73-172 $mm^3$ were divided into 3 groups randomly. Then mice were treated by in situ injection for eleven days with 0.4 µmol PG and PG-iron complex, respectively, and water was used as control. Tumor size was measured with a caliper every other day and calculated by the formula, tumor volume = shortest $diameter^2 \times$ longest diameter/2. All mice were sacrificed after 11-day treatment.

Fresh tumor tissues were fixed with 4% paraformaldehyde, washed with PBS, and stored at 4 °C. The tissues were dehydrated and embedded in paraffin as previously described[69,70], and were sectioned at a thickness of 4 µm. Perls prussian blue and Nuclear Fast Red staining was carried out as previously described[71]. Immunohistochemistry analysis was performed as previously described[69,70] using the anti-GPX4 and anti-4-HNE antibodies (Abcam ab125066, 1: 200 diluted and ab48506, 1: 50 diluted) and DAB Horseradish Peroxidase Color Development Kit (Beyotime, China), and cell nucleus was strained using hematoxylin. The images were obtained using Nikon Eclipse E100 microscope, and quantified using ImageJ with a plug-in tool IHC Profiler. The animal experiments were performed according to the standards set forth in the Guide for the Care and Use of Laboratory Animals (National Institutes of Health, 1985). Experimental protocols were approved by the Animal Care Committee at College of Life Sciences of Shandong University (SYDWLL-2023-014).

## Biosynthesis of PG

The strains Q3595 and Q4333 were grown overnight in LB broth and 1:100 diluted into 250 mL shaking flasks with 50 mL of MSM with and without iron, respectively. After incubation at 37 °C, 0.1 mM IPTG was added for induction at OD600 of 0.8, and the temperature was decreased to 30 °C for further cultivation of 12 h. PG concentration was determined from the 446 nm value using the colorimetric reaction between PG and cinnamaldehyde. Enzymatic activity of fumarase and aconitase was measured using Fumarase Activity Colorimetric Assay Kit (Suzhou Grace Biotechnology, China) and Aconitase Activity Assay Kit (Solarbio, China).

## Phenol biodegradation

Phenol degradation was carried out using M9 minimal medium containing 1 g/L phenol, 4.36 g/L $K_2HPO_4$, 4.485 g/L $NaH_2PO_4 \cdot 2H_2O$, 1 g/L $NH_4Cl$, 1.058 g/L $MgSO_4 \cdot 7H_2O$, and 1 mL of trace element solution (36 g/L $CaCl_2$,

3.7 g/L $CoCl_2 \cdot 6H_2O$, 1 g/L $MnCl_2 \cdot 4H_2O$, 0.2 g/L $Na_2MoO_4 \cdot 2H_2O$). For iron depleted- and iron rich-conditions, 3.7 mg/L and 0.5 g/L $FeSO_4 \cdot 7H_2O$ was supplemented in the M9 medium. *Pseudomonas* sp. DHS3Y was grown overnight in LB broth, diluted 1:50 into M9 medium, and grown at 32 °C. Concentration of residue phenol was determined as previously dicribed[72].

## Statistics and reproducibility

The strain growth assay, phenols challenge experiments, enzyme activity assay, phloroglucinol biosynthesis, phenol degradation, decolorization of malachite green were performed at least three replicates. Three biological independent samples with two technical repeats for each sample were used for qRT-PCR. Seven biological independent samples were used for in vivo tumor growth and treatment. Sample size was determined based on the previous experience and the majority of other metabolic engineering publications. It is sufficient to confirm that results did not vary and were consistent. Data were analyzed using Microsoft Excel 2019 and plotted using Graphpad Prism 10 (Graphpad). Two-tailed Student's tests and two-way ANOVA assays were performed to determine the statistical significance.

## Reporting summary

Further information on research design is available in the Nature Portfolio Reporting Summary linked to this article.

## Data availability

All source data underlying the graphs and charts presented in the main figures are provided in Supplementary Data 1. The uncropped images in Western blot were shown in Supplementary Fig. 5 and 6. The strains and plasmids used in this study were provided in Supplementary Information, and primers used in this study were provided in Supplementary Data 2. The raw data of genome resequencing were submitted to NCBI with the Accession number of PRJNA1063523.

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

## Acknowledgements
We thank Prof. Yin Li and Zhen Cai (Institute of Microbiology, Chinese Academy of Sciences) for pQ-lib, Prof. Sheng Yang (Institute of Plant Physiology and Ecology, Chinese Academy of Sciences) for plasmids pTargetF, pCas, pKSI-1, pREDTKI, and pMDIAI, Prof. Zhiqi Cong and Dexin Feng (Qingdao Institute of Bioenergy and Bioprocess Technology) for helpful discussion, and Pengyu Zhao for Fig. 7a. This study was financially supported by the NSFC (32170085, 32370068), National Key Research and Development Program of China (2022YFC2104700), the Fundamental Research Funds for the Central Universities (2021JCG025), Distinguished Scholars Program of Shandong University (G.Z.), and State Key Laboratory of Microbial Technology (WZCX2021-02, SKLMTFCP-2023-03).

## Author contributions
G.Z. designed the experiments. X.S., J.W., Z.Z., Min Liu, Miaomiao Liu, Y.F., C.S., and X.F. performed the experiments. G.Z., X.S., J.W., B.L., D.S., S.L., Q.Q., and M.X. analyzed the results. G.Z. and X.S. wrote the manuscript. All authors edited the manuscript before submission.

## Competing interests
The authors declared no competing interests.
