## [Peer Review File · Communications Biology]

Reviewers' comments:

Reviewer #1 (Remarks to the Author):

The authors Xinyue Sui and co-workers submitted their manuscript entitled “Phenolic compounds induce ferroptosis by promoting the hydroxyl radical generation in Fenton reaction” to the journal “Communications Biology” in order to be considered for publication as an article.

The research work deals with investigation of phenols that induce ferroptosis in the presence of iron ions due to the formed metal complexes. In addition, three genes/operon are described to contribute to tolerance towards phenols, i.e. *sodB*, *fetAB*, and *ClpXP*. The findings contribute to the understanding of phenol toxicity (in the presence of iron).

However, the induction of ferroptosis by iron-containing complexes is already described in literature. Not in particular for phenols but on the examples of chlorido[N,N'-disalicylidene-1,2-phenylenediamine]iron(III) complexes, where complexation of iron(III) also takes place by phenolic groups. Unfortunately, the authors do merely slightly focus on antioxidative properties known of phenols. There are even reports that polyphenols can inhibit ferroptosis (Srai et al., Antioxidants). The authors have hardly included a discussion of such aspects in their manuscript. This should definitely be included in order to get a holistic picture. Moreover, please see the following major comments:

Lines 167-169: “The concentration of iron in medium used in ALE was much higher than those of manganese, copper and zinc...” Please provide the particular values of the content of these minerals. Which method was employed for quantification? AAS, ICP-MS or something else? What would it look like in the opposite case with an iron-deficient medium that has been treated with an iron-specific chelator (not affecting manganese, copper, zinc), for example? Such investigations could provide further valuable information about the significance of iron.

Lines 213/271: “Subsequently, iron-free medium was used” / “iron-free minimal medium” How was it ensured that the medium really does not contain any iron? The authors are asked to provide more detailed information on this. Also line 326: “iron-containing medium”. How was the iron content assured?

@ Figure 2b, 2f: How about the combination of H₂O₂+Fe for comparison respectively for verification of toxic effects? The authors are kindly asked to add such data/experiment. Moreover, Fe is understood to be elemental iron. However, this was probably not used but ionic iron. Was it Fe²⁺ or Fe³⁺ (the latter as stated in Figure 2g)? The authors are asked to be more precise in this regard – also Figure 3d, 3e; Figure 5f, 5g, 5h, 5i, Figure 6.

Lines 225-...: “majority of iron inside the cells forms complex... so, the Fenton reaction is not significant” As far as I understood, it was generally reported by Gust et al. J Med Chem before that intact iron complexes are also potent to induce ferroptosis via the Fenton reaction. What is the authors' explanation that the Fenton reaction should not play a role here in their particular case? Please discuss in the manuscript.

Lines 276-277: “generation of HO[•] promoted by PG-iron complex was the main factor of PG toxicity to E. coli” This finding is somehow in contrast to that of Gust et al. Eur J Med Chem who found that Gram-negative bacteria such as E. coli may be less susceptible to oxidative-ferroptotic cell death caused by iron complexes due to different structure of the cell wall compared to Gram-positive. What explanation do the authors have for the fact that their PG-iron complex is toxic? Please discuss in the manuscript.

Lines 376-377: “ferroptotic cell death pathway in diverse organisms, from bacteria to mammalian cells” also Lines 564-565: “We demonstrate that phenol-iron complex induces ferroptosis in diverse organisms, which has great application potential in different fields” I guess this is already known, e.g., 10.1084/jem.20181776, <https://doi.org/10.4014/jmb.2307.07002>, <https://doi.org/10.1186/s12935-021-02366-0>, among many others.

Lines 481-482: “which phenolic compounds can induce ferroptotic cell death”... well, iron ions are required to induce ferroptosis; therefore, it is not the phenols themselves but the phenolic iron complexes.

Line 606: “phenolic compounds are highly toxic to human” That is true, but would not be of such great importance when killing tumor cells in the treatment of cancer. However, use of phenols for the treatment of bacterial infections seems less valuable, also considering the carcinogenic potential of phenols (e.g., as reviewed by Stich, Mutat Res). Thus, the range of application in the healthcare section seems limited.

Minor comments (this list is not intended to be exhaustive):

Lines 51-52: Are there more recent data available than from 2015?

Lines 78-79: superoxide dismutase = “SOD” instead of “SODs”?

Line 89: “it is” instead of “it’s”

Lines 118/121/128/183/185/187/190/348/351/354: “treatment with PG” instead of “PG challenge”

Line 192: 1.3 instead of 1. 3

Lines 223-224: “required the millimolar concentration of H₂O₂” Please provide a particular range of concentration.

Line 245: “acetonitrile-d₃” please use subscript d₃

Line 248: “scavenger t-butanol” please use in italics t-

Line 294: “iron treatment” instead of “iron challenge” – moreover, which iron: Fe(II) or Fe(III)?

Line 309: “Western blot” instead of “western blot”; kind of personal name, named in reference to the Southern blot (Edwin Southern)

Line 322: “ferroptosis... cell death that was recently described” Well, the term ferroptosis was coined/described as early as 2012 by Dixon et al., Cell – one decade ago.

Line 341: “induced the ferroptosis-like cell death” Why do the authors just say “ferroptosis-like” and do not call it ferroptosis? The criteria of ferroptosis are met, aren’t they?

Line 360: “n” in italics

Figure 5g: “K. pneumoniae” instead of “Klebsiella” and “S. typhimurium” instead of “Salmonella”.

Line 358/371/644: “Typhimurium” lower case and in italics.

Line 368: "could not" instead of "couldn't"

Line 408: "was not" instead of "wasn't"

Figure 7d: Scale bar is hard to read. Please improve.

Figure 8: Very nice, well done!

Line 489: "5c" instead of "5C"

Line 534: "we would" instead of "we'd"

Line 589: Please provide newer data, e.g., <https://doi.org/10.3322/caac.21763>

Line 736/737/749: "37 °C" instead of "37°C" / "30 °C" instead of "30°C" / ...

Lines 744-747: use of space when reporting the hydrate-containing chemicals

All the best!

Reviewer #2 (Remarks to the Author):

The manuscript titled "Phenolic compounds induce ferroptosis by promoting the hydroxyl radical generation in Fenton reaction" delves into the realm of phenolic compound toxicity, utilizing *E. coli* and phloroglucinol to unravel the underlying molecular mechanisms. This study is pivotal in uncovering how these compounds trigger ferroptotic cell death, a type of iron and oxygen-dependent cell death, across a range of organisms. The researchers have meticulously identified genetic mutations in *E. coli* that impart phenol tolerance and have showcased the potential applications in biotechnology and healthcare. The novelty of this research lies in its exploration of ferroptosis induction by phenolic compounds, a previously uncharacterized pathway, offering new insights into cell death mechanisms and their broader implications.

Detailed Evaluation

The manuscript provides a comprehensive introduction, effectively setting the stage by emphasizing the industrial significance and environmental impact of phenolic compounds. It addresses the gap in understanding their toxicity, which is crucial for refining biochemical processes involving these compounds. In terms of originality, the study marks a significant advancement in the field by elucidating the mechanism through which phenolic compounds induce ferroptosis. This finding is not only novel but also fills a critical gap in the existing literature on phenolic toxicity. The methodology employed, particularly the use of adaptive laboratory evolution to identify *E. coli* genes linked to phenol tolerance, is robust and well-executed. The combination of *in vitro* and *in vivo* experiments lends credibility and depth to the study's conclusions. The impact of this research is far-reaching, with implications across diverse fields, including biotechnology and medicine. The revelation that phenolic compounds can induce ferroptosis-like cell death paves the way for new therapeutic and biotechnological applications. The manuscript is well-structured, with clear and coherent presentation of findings. Integrating the results with existing knowledge, the discussion highlights the significance and potential applications of these discoveries. While the statistical methods appear adequate, a more detailed description and availability of raw data would enhance the ability to fully assess the robustness and reproducibility of the findings.

Errors and Suggestions:

- Lines 102 & 103: Greater clarity in describing the concentrations used and observed results would enhance understanding.
- Lines 165-173: A more detailed explanation of how genes *sodB* and *fetAB* affect phenol tolerance and their relation to oxidative stress would be beneficial.
- Lines 279-288: Expanding on the role of the *clpX* gene and its relation to RpoS protein stability would add depth to its significance in the study.
- Lines 319-329: Additional details on assays used to confirm ferroptosis induction by phenolic compounds are recommended.
- General: Clarification is needed in sections discussing the statistical analyses and methodologies to ensure reproducibility.
- Figures and Tables: Some figures and tables could benefit from more detailed legends and explanations to improve their interpretability.
- References: Checking for the latest references and ensuring all cited works are up-to-date would strengthen the manuscript's relevance.
- Technical Language: Simplifying complex jargon where possible could make the paper more accessible to a broader audience.
- Analysis of Figures and Results: The figures in the manuscript effectively illustrate the key findings, especially the impact of phenolic compounds on ferroptosis in *E. coli*. However, some figures could be improved for clarity. For instance, the graphical representations of the molecular pathways involved in ferroptosis could be more detailed to enhance understanding. The results showing the induction of ferroptosis in various organisms highlight the broad applicability of the findings, yet a more thorough discussion of these results in the context of existing literature would be beneficial.

In conclusion, the manuscript is a valuable contribution to the field, offering novel insights into the mechanism of ferroptosis induction by phenolic compounds. While the study is well-conducted and presents significant findings, the implementation of the suggested corrections and improvements, particularly in enhancing figure clarity, expanding methodological details, and providing a more thorough discussion of results, would strengthen the paper. With these revisions, the paper is potentially acceptable for

Reviewer #3 (Remarks to the Author):

The current work by Sui et al. aims to uncover the mechanism of phloroglucinol (PG) toxicity in *E. coli*. The authors used different genetic and biochemical methods to investigate this matter. The experiments are well-designed, and the paper is written in a cohesive and logical manner. I believe this work demonstrated the mechanism of PG toxicity beautifully. However, I do not think it is a form of ferroptosis:

A- Ferroptosis is reported as the overwhelming iron-dependent accumulation of lethal lipid ROS (PMID 22632970).

B- Bacterial cell death due to the interaction of Fe and ROS has been documented for decades.

C- Lipid peroxidation is not a major part of oxidative stress in bacteria, because it needs PUFAs that are not present in bacteria (except for the thylakoid membranes of photosynthetic bacteria).

Based on these points, one would be hesitant to call the cell death via PG “ferroptosis”, but another example of cell death due to interplay of Fe and ROS. On a separate note, an alternative mechanism for the role of ClpXP in the process is suggested below, and it is worth investigating.

My major and minor comments are as follows:

Major:

1- While the authors mention the discrepancy between their results and the previous reports regarding SODs (they have been reported to get induced by phenolic compounds, and here deletion of *sodB* protects the cells), they do not attempt to explain it. Per the results of this work, induction of SODs in presence of phenolic compounds would result in production of more peroxide and hence more cell death. So why do cells induce SOD under these conditions? While pinpointing the exact reason may be hard without further investigation, it would be worth mentioning a few plausible scenarios for this conflict with previous reports.

2- The mutants should be complemented.

3- Per the presented model, one would expect $\Delta katG$ and or $\Delta katE$ mutants to be more susceptible to PG, but only the overexpression data is provided in figure 2a. Testing these strains could strengthen the argument.

4- In the PG experiments where potentially toxic compounds are tested (for example Fe in 2d) or when important genes are deleted (like *rpoS* in 4b), it is essential to have a no-PG control for every condition. One could assume that deleting *rpoS* could affect cell growth and survival by itself. So, a $\Delta rpoS$ + PG must be compared to both WT + PG and $\Delta rpoS$ no PG. Regarding Fe, it is shown in 2f that Fe doesn't change the survival, however it's not mentioned what concentration of Fe is used here and no comparison can be made to 2g. Please refer to minor point #4.

5- It was clearly shown that ClpXP+*rssB* degrade RpoS. However, the role of RpoS in the process is still not clear. Throughout the manuscript, it was mentioned that RpoS is a major regulator of ROS response. While it does induce some ROS response genes, the major regulators during log phase (when these experiments are performed) are OxyR and SoxRS, and these two regulate the most important ROS response factors.

6- Relevant to point#5, recent work by Sen and Imlay (PMID 32601069) suggests that ClpXP degrades DPS and results in the release of iron. This fits beautifully to the data presented here: the $\Delta clpXP$ mutants have more intact DPS, and less free iron to react either PG and cause stress and death. Based on the presented data here and the published data, it seems more plausible that the effect of ClpXP in the process is through DPS and not (only) RpoS. I would highly recommend the authors to test this alternative hypothesis.

Minor:

1- Lines 91 to 97 are repeats of lines 34 to 40.

2- The figures throughout the paper should be properly labeled. For example, fig 2a and 2b should have “WT” and “ $\Delta sodB$ ” labels. Also, the all the assay components should be mentioned. For example, the addition of PG is mentioned in 2b and 2f, but not in the rest of the panels.

- 3- A few typos and grammatical errors were present. Proofreading would be helpful.
- 4- How much Fe is present in figure 2f?
- 5- What Fe containing compound is used?
- 6- Figure 4C: a control band of a housekeeping protein should be presented. Quantifying the data by image analysis would also be helpful.
- 7- Lines 558-9: RpoS is clearly visible in the WT strain, albeit at a lower amount.
- 8- KatE is probably the most important part of the RpoS regulon in response to ROS, and its mRNA amount should be presented in Figure 4d.
- 9- The staining data in figure 7d needs some sort of quantification. It appears that the staining level is the same in PG and PG+Fe with the GPX4.
- 10- 10- No survival data is reported for figure 7.

Ref: COMMSBIO-23-4240

Phenolic compounds induce ferroptosis by promoting the hydroxyl radical generation in Fenton reaction

Response to the reviewers' comments:

We thank the reviewers for their constructive comments and appreciation for the importance of our study: "*The findings contribute to the understanding of phenol toxicity.*" (Reviewer 1); "*the manuscript is a valuable contribution to the field, offering novel insights into the mechanism of ferroptosis induction by phenolic compounds.*" (Reviewer 2); "*this work demonstrated the mechanism of PG toxicity beautifully.*" (Reviewer 3).

We responded to the Reviewers' comments in a point-by-point form.

We appreciate the opportunity to re-submit our study to Communications Biology and are happy to make any additional modifications suggested by the Editors and Reviewers.

Reviewer #1

1. the induction of ferroptosis by iron-containing complexes is already described in literature. Not in particular for phenols but on the examples of chlorido[N,N'-disalicylidene-1,2-phenylenediamine]iron(III) complexes, where complexation of iron(III) also takes place by phenolic groups. Unfortunately, the authors do merely slightly focus on antioxidative properties known of phenols. There are even reports that polyphenols can inhibit ferroptosis (Srai et al., Antioxidants). The authors have hardly included a discussion of such aspects in their manuscript. This should definitely be included in order to get a holistic picture.

Reply: Thanks for the information. We have introduced that the salen/salophene iron complexes can stimulate HO· production in Fenton reaction and induce ferroptosis in the revised manuscript (page 10, line 195-197; page 23, line 464-466).

In this study, we have tested several simple phenolic compounds and they can induce ferroptosis in diverse organisms. In contrast, polyphenols, natural products commonly found in higher plants, show antioxidant activity and inhibit ferroptosis. This divergence is probably due to the molecular structure differences between simple phenolic compounds and polyphenols which usually contains several benzene rings and different functional groups. This was discussed in the revised manuscript (page 20, line 407-413).

2. Lines 167-169: “The concentration of iron in medium used in ALE was much higher than those of manganese, copper and zinc...” Please provide the particular values of the content of these minerals. Which method was employed for quantification? AAS, ICP-MS or something else? What would it look like in the opposite case with an iron-deficient medium that has been treated with an iron-specific chelator (not affecting manganese, copper, zinc), for example? Such investigations could provide further valuable information about the significance of iron.

Reply: In previous study, we constructed engineered *E. coli* strain to produce PG from glucose, and carried out fermentation using a defined minimal salt medium, which contains 1.15 mM ferric ammonium citrate, 10 μ M ZnSO₄, 10 μ M CuSO₄, and 80 μ M MnCl₂. In this study, this defined medium was used in the adaptive evolution to obtain PG-tolerant strains. All the metal salts were quantitatively added into the medium, and the concentration is not measured.

As iron is necessary for PG toxicity and PG can not kill *E. coli* in iron-deficient medium (Fig. 2c), we can not evolve PG-tolerant strain under iron-deficient condition and that sentence was deleted in the revised manuscript.

3. Lines 213/271: “Subsequently, iron-free medium was used” / ” iron-free minimal medium” How was it ensured that the medium really does not contain any iron? The authors are asked to provide more detailed information on this. Also line 326: “iron-containing medium”. How was the iron content assured?

Reply: In this study, a defined minimal salt medium was used, and iron was added into the medium quantitatively. (page 25, line 512-517)

4. @ Figure 2b, 2f: How about the combination of H₂O₂+Fe for comparison respectively for verification of toxic effects? The authors are kindly asked to add such data/experiment. Moreover, Fe is understood to be elemental iron. However, this was probably not used but ionic iron. Was it Fe²⁺ or Fe³⁺ (the latter as stated in Figure 2g)? The authors are asked to be more precise in this regard – also Figure 3d, 3e; Figure 5f, 5g, 5h, 5i, Figure 6.

Reply: When we carried out the experiment shown in Fig. 2g (previous Fig. 2c), we used the defined medium that contains 1.15 mM ferric ion and was used in PG bioproduction and adaptive evolution. That is, all data in Fig. 2g were generated with the presence of iron. We have added the concentrations of Fe³⁺, H₂O₂, and PG in Fig. 2g and other related figures.

In all experiments testing bacterial survival after treatment with PG, Fe³⁺ was used, and this information was added in all related figures in the revised manuscript.

5. Lines 225-...: “majority of iron inside the cells forms complex... so, the Fenton reaction is not significant” As far as I understood, it was generally reported by Gust et al. J Med Chem before that intact iron complexes are also potent to induce ferroptosis via the Fenton reaction. What is the authors' explanation that the Fenton reaction should

not play a role here in their particular case? Please discuss in the manuscript.

Reply: Gust et al. have reported a series of iron-containing complex t including Chlorido[N,N'-disalicylidene-1,2-phenylenediamine]iron(III) Complexes (J Med Chem, 2019, 62: 8053) and salen/salophene iron complexes (Eur J Med Chem, 202, 209: 112907). This kind of complexes composed of iron and small molecules, just like PG-iron complex reported here, can induce ferroptosis via the Fenton reaction in the form of intact complex. In our manuscript, we are discussing the function of iron-storage proteins FtnA, Bfr, and Dps. These proteins can sequester iron and limit the potential for iron-dependent HO· formation. (page 10, line 191-194)

6. Lines 276-277: “generation of HO· promoted by PG-iron complex was the main factor of PG toxicity to *E. coli*” This finding is somehow in contrast to that of Gust et al. Eur J Med Chem who found that Gram-negative bacteria such as *E. coli* may be less susceptible to oxidative-ferroptotic cell death caused by iron complexes due to different structure of the cell wall compared to Gram-positive. What explanation do the authors have for the fact that their PG-iron complex is toxic? Please discuss in the manuscript.

Reply: We claimed that generation of HO· promoted by PG-iron complex was the main factor of PG toxicity to *E. coli* based on the following facts: (1) iron is necessary for the PG toxicity and PG can not kill *E. coli* under iron-deficient condition (Fig. 2f-h); (2) the intracellular HO· concentration increased significantly only with the presence of both PG and iron (Fig. 3e); (3) the HO· scavenger dimethyl sulfoxide rescued *E. coli* cells from the PG-iron killing assay (Fig. 3f).

We don't think that this result is in contrast to above mentioned paper published by Gust et al. (Eur J Med Chem, 209: 112907). In these two studies, the susceptibility testing was carried out under different conditions. In that paper, the iron complexes were tested in a concentration range covering the serial dilutions from 100 to 0.391 µg/mL. In our study, 1.3 g/L PG was used to mimic the PG bioproduction process. It is very possible that the iron complexes in Gust's paper can kill *E. coli* at a much higher concentration. Discussion was

added in the revised paper. (page 15, line 297-299)

7. Lines 376-377: “ferroptotic cell death pathway in diverse organisms, from bacteria to mammalian cells” also Lines 564-565: ”We demonstrate that phenol-iron complex induces ferroptosis in diverse organisms, which has great application potential in different fields” I guess this is already known, e.g., 10.1084/jem.20181776, <https://doi.org/10.4014/jmb.2307.07002>, <https://doi.org/10.1186/s12935-021-02366-0>, among many others.

Reply: Indeed, as you mentioned, it has been known and reported in previous literatures that ferroptosis occurs in diverse organisms. Here, what we want to emphasize is that phenol-iron complex can induce ferroptosis, different from previous discovered ferroptosis inducer. The related sentence has been revised (page 23, line 464-466).

8. Lines 481-482: “which phenolic compounds can induce ferroptotic cell death”... well, iron ions are required to induce ferroptosis; therefore, it is not the phenoles themselves but the phenolic iron complexes.

Reply: This sentence has been revised to “phenolic compounds can form complex with iron ions and induce ferroptotic cell death”. (page 19, line 377-378)

9. Line 606: “phenolic compounds are highly toxic to human” That is true, but would not be of such great importance when killing tumor cells in the treatment of cancer. However, use of phenols for the treatment of bacterial infections seems less valuable, also considering the carcinogenic potential of phenols (e.g., as reviewed by Stich, *Mutat Res*). Thus, the range of application in the healthcare section seems limited.

Reply: Thanks for your comments. Based on your suggestion, the part discussing potential applications in the healthcare field was simplified in the revised manuscript.

10. Lines 51-52: Are there more recent data available than from 2015?

Reply: We have updated the annual production of phenol according to a paper published in 2023 (page 3, line 46-47).

11. Lines 78-79: superoxide dismutase = “SOD” instead of “SODs”?

Reply: It has been corrected accordingly (page 4, line 74).

12. Line 89: “it is” instead of “it’s”

Reply: It has been corrected accordingly (page 5, line 84).

13. Lines 118/121/128/183/185/187/190/348/351/354: “treatment with PG” instead of “PG challenge”

Reply: It has been corrected accordingly (page 7, line 134; page 8, line 152; page 14, line 278; page 21, line 418; page 42, line 865/868/878; page 43, line 890/896; page 44, line 920; page 45, line 931/934/940).

14. Line 192: 1.3 instead of 1. 3

Reply: It has been corrected accordingly (page 43, line 888).

15. Lines 223-224: “required the millimolar concentration of H₂O₂” Please provide a particular range of concentration.

Reply: *In vivo* experiments in *E. coli* suggested that DNA damage required H₂O₂ at a concentration of 1 to 2 mM. This has been added in the revised manuscript (page 10, line 187-190).

16. Line 245: “acetonitrile-d₃” please use subscript d₃

Reply: It has been corrected accordingly (page 44, line 905).

17. Line 248: “scavenger t-butanol” please use in italics t-

Reply: It has been corrected accordingly (page 44, line 908).

18. Line 294: “iron treatment” instead of “iron challenge” – moreover, which iron: Fe(II) or Fe(III)?

Reply: It has been corrected accordingly, and ferrous ion was used in this experiment (page 44, line 920).

19. Line 309: “Western blot” instead of “western blot”; kind of personal name, named in reference to the Southern blot (Edwin Southern)

Reply: It has been corrected accordingly (page 12, line 249; page 28, line 580).

20. Line 322: “ferroptosis... cell death that was recently described” Well, the term ferroptosis was coined/described as early as 2012 by Dixon et al., Cell – one decade ago.

Reply: the word “recently” in this sentence has been removed (page 13, line 270).

21. Line 341: “induced the ferroptosis-like cell death” Why do the authors just say “ferroptosis-like” and do not call it ferroptosis? The criteria of ferroptosis are met, aren't they?

Reply: Yes, our results showed that the criteria of ferroptosis were met. As ferroptosis was coined in eukaryotic organisms, some previous papers with results from bacteria used the phrase “ferroptosis-like cell death”. To avoid misunderstanding, we used the word “ferroptosis” throughout in the revised manuscript.

22. Line 360: "n" in italics

Reply: It has been corrected accordingly (page 46, line 946).

23. Figure 5g: "K. pneumoniae" instead of "Klebsiella" and "S. typhimurium" instead of "Salmonella".

Reply: It has been corrected accordingly (page 45, line 944-945).

24. Line 358/371/644: "Typhimurium" lower case and in italics.

Reply: It has been corrected accordingly (page 15, line 296; page 26, line 534).

25. Line 368: "could not" instead of "couldn't"

Reply: It has been corrected accordingly (page 15, line 293).

26. Line 408: "was not" instead of "wasn't"

Reply: It has been corrected accordingly (page 16, line 320).

27. Figure 7d: Scale bar is hard to read. Please improve.

Reply: It has been corrected accordingly (Fig. 7d).

28. Figure 8: Very nice, well done!

Reply: Thanks.

29. Line 489: “5c” instead of “5C”

Reply: It has been corrected accordingly (page 19, line 385).

30. Line 534: “we would” instead of “we’d”

Reply: It has been corrected accordingly (page 21, line 418).

31. Line 589: Please provide newer data, e.g., <https://doi.org/10.3322/caac.21763>

Reply: The number of cancer death in the United States in 2023 estimated by the American Cancer Society was used in the revised manuscript (page 24, line 490).

32. Line 736/737/749: “37 °C” instead of “37°C” / “30 °C” instead of “30°C” / ...

Reply: It has been corrected accordingly (page 26, line 545; page 30, line 632/633).

33. Lines 744-747: use of space when reporting the hydrate-containing chemicals

Reply: It has been corrected accordingly (page 25, line 513-517; page 31, line 641-643).

Reviewer #2

1. Lines 102 & 103: Greater clarity in describing the concentrations used and observed results would enhance understanding.

Reply: The medium and PG concentrations used in the killing assay were

added in the revised manuscript (page 5, line 96-100).

2. Lines 165-173: A more detailed explanation of how genes *sodB* and *fetAB* affect phenol tolerance and their relation to oxidative stress would be beneficial.

Reply: In this part, we introduced some general background information about SOD, and how *sodB* and *fetAB* affect PG tolerance is not involved here. The effect of FetAB and SOD on *E. coli* tolerance was added after the description of related results in the revised manuscript (page 8, line 158-159; page 9, line 184).

3. Lines 279-288: Expanding on the role of the *clpX* gene and its relation to RpoS protein stability would add depth to its significance in the study.

Reply: ClpXP is an ATP-dependent protease complex involved in numerous biological processes. In the complex, ClpP subunits form the proteolytic center and ClpX serves as a substrate-specifying adaptor to unfold and transfer the substrate protein to the catalytic site. RpoS is the major regulator of general stress response in *E. coli*. In growing cells, RpoS has an extremely short half-life due to its proteolysis by the ClpXP protease, and various stress conditions can lead to stabilization and accumulation of RpoS. In addition, the proteolysis of RpoS requires a direct recognition factor RssB which has specific affinity for RpoS and targets RpoS to the ClpXP protease. Following your suggestion, we have added some new information about ClpXP and RpoS here (page 12, line 234-239).

4. Lines 319-329: Additional details on assays used to confirm ferroptosis induction by phenolic compounds are recommended.

Reply: As the main text is limited to 5000 words, some experimental details were added in the section of "Methods" (page 26, line 526-531).

5. General: Clarification is needed in sections discussing the statistical analyses and methodologies to ensure reproducibility.

Reply: In the revised manuscript, we added a section of “Statistics and Reproducibility” to describe general information on how the analyses of the data were conducted, and general information on the reproducibility of experiments. (page 31, line 648-658)

In the section of “Methods”, some detail information was added, including the components and corresponding concentration used in killing assay (page 26, line 526-531), the antibody used to detect His6 tagged protein (page 28, line 586), software used to quantify the Western blot bond (page 28, line 588) and IHC results (page 30, line 623), and mice sacrifice (page 30, line 613).

6. Figures and Tables: Some figures and tables could benefit from more detailed legends and explanations to improve their interpretability.

Reply: For clearer presentation, more information, including strains, overexpressed genes, and components used in the killing assay, were labeled in related figures in the revised manuscript.

7. References: Checking for the latest references and ensuring all cited works are up-to-date would strengthen the manuscript's relevance.

Reply: Thanks for the suggestion, and some references were updated. For example, the annual production of phenol in 2023 and the number of cancer death in the United States in 2023 were used in the revised manuscript (Ref. #2 and Ref. #62).

8. Technical Language: Simplifying complex jargon where possible could make the paper more accessible to a broader audience.

Reply: Thanks for your suggestion. The manuscript has been carefully reviewed and revised throughout.

9. Analysis of Figures and Results: The figures in the manuscript effectively illustrate the key findings, especially the impact of phenolic compounds on ferroptosis in *E. coli*. However, some figures could be improved for clarity. For instance, the graphical representations of the molecular pathways involved in ferroptosis could be more detailed to enhance understanding. The results showing the induction of ferroptosis in various organisms highlight the broad applicability of the findings, yet a more thorough discussion of these results in the context of existing literature would be beneficial.

Reply: For clearer presentation, more information, including strains, overexpressed genes, and components used in the killing assay, were labeled in related figures in the revised manuscript.

In fig. 8 showing the molecular pathways involved in ferroptosis, we added the following information: (1) inactivation of ClpXP increased the protein level of Dps, that will help to reduce the intracellular concentration of free iron. (2) KatE is an important member of RpoS regulon and plays a critical role in hydrogen peroxide elimination.

In the section of ferroptosis in various organisms, we discussed the induction of ferroptosis in gram-positive bacteria (page 15, line 297-299). Although we have simplified some sections, due to the length limitation, not all the issues raised by the reviewers can be discussed in detail. Sorry about that.

Reviewer #3

1. I believe this work demonstrated the mechanism of PG toxicity beautifully. However, I do not think it is a form of ferroptosis:

A- Ferroptosis is reported as the overwhelming iron-dependent accumulation of lethal lipid ROS (PMID 22632970).

B- Bacterial cell death due to the interaction of Fe and ROS has been documented for decades.

C- Lipid peroxidation is not a major part of oxidative stress in bacteria, because it needs PUFAs that are not present in bacteria (except for the thylakoid membranes of photosynthetic bacteria).

Based on these points, one would be hesitant to call the cell death via PG “ferroptosis”, but another example of cell death due to interplay of Fe and ROS. On a separate note, an alternative mechanism for the role of ClpXP in the process is suggested below, and it is worth investigating.

Reply: Thanks for your comments. As you mentioned above, ferroptosis is characterized by the overwhelming, iron-dependent accumulation of lethal lipid ROS (Cell, 149: 1060). Although the term ferroptosis was coined in 2012, ferroptosis-like cell death were actually observed long before. For instance, in the 1950s and 1960s, Harry Eagle demonstrated that depletion of the amino acid cysteine can cause cell death and that the endogenous synthesis of cysteine makes cells resistant to such cell death (Nat Rev Mol Cell Biol, 22: 267). Now, we know that cysteine is rate-limiting for the biosynthesis of GSH and that sustaining GSH synthesis or promoting the activity of GSH peroxidase 4 can protect cells from ferroptosis, linking those early studies to the mechanism of ferroptosis. So, we don't think that we can rule out that bacterial cell death caused by PG-iron complex is ferroptosis because bacterial death due to the interaction of iron and ROS has been reported.

Although PUFAs are not present in bacteria, bacterial ferroptosis or ferroptosis-like death has been reported by different groups in the past few years, indicating that the bacterial ferroptotic cell death has been widely recognized. Gust et al. discovered that salen/salophene metal complexes induced ferroptosis of methicillin-resistant *Staphylococcus aureus* (MRSA), a formidable pathogen whose emergence and prevalence have increased the threat of acute pneumonia (Eur J Med Chem, 2021, 209: 112907). The reported MRSA

ferroptosis inducers also include sodium alginate-Fe₃O₄-cinnamaldehyde particles coated with erythrocyte membrane and platelet membrane (ACS Nano, 2023, 17: 11692), Ferrous sulfate-loaded hydrogel (Biomaterials, 2022, 290: 121842), nano-decocted ferrous polysulfide (Nano Today, 2020, 35: 100981), monatomic transition metal sites anchored on sp²c-linked covalent organic frameworks (Adv Sci, 2023, 10: 2207507), and Fe-doped titanite (ACS Nano, 2023,17: 2711). Wang et al. reported that 5-aminolevulinic acid photodynamic therapy promotes ferroptosis-like death of *Mycobacterium abscessus* (Antioxidants, 2022, 11: 546). Ma et al. reported the ferroptotic death of *E. coli*, *S. aureus* and *Salmonella pullorum* in the macrophage cells (Theranostics, 2022, 12: 2266). All these bacterial cell death share the characteristics of ferroptosis, including ROS production, lipid peroxidation, and GSH depletion, consistent with our results.

Following your suggestion, the role of ClpXP was further demonstrated. We proved that *clpX* deletion increased the protein stability of RpoS and Dps. Then, RpoS enhanced the transcription level of several oxidative stress response genes including *katE* whose overexpression protected *E. coli* from PG toxicity. On the other hand, Dps is a miniferritin and sequesters free iron, and the increased Dps protein would help to reduce the intracellular concentration of free iron and production of hydroxyl radicals. These results were added in the revised manuscript (page 15, line 258-263).

2. While the authors mention the discrepancy between their results and the previous reports regarding SODs (they have been reported to get induced by phenolic compounds, and here deletion of *sodB* protects the cells), they do not attempt to explain it. Per the results of this work, induction of SODs in presence of phenolic compounds would result in production of more peroxide and hence more cell death. So why do cells induce SOD under these conditions? While pinpointing the exact reason may be hard without further investigation, it would be worth mentioning a few plausible scenarios for this conflict with previous reports.

Reply: In some microorganisms, the SODs are found to be upregulated after treatment with phenol, and it was believed that SODs contributed to resistance

of phenols-induced oxidative stress as they are members of antioxidant enzymes. In *E. coli*, transcription of *sodA* gene is upregulated by SoxRS, principal regulator of the superoxide radical response (Cell Reports, 12: 1289), and the *sodB* gene is positively regulated by Fur through a small RNA RyhB (PNAS, 99: 4620). So, it is very likely that the SOD genes are activated upon exposure to PG and iron. This is discussed in the revised manuscript (page 21, line 426-429).

3. The mutants should be complemented.

Reply: Based on your suggestion, the function of *clpX*, *sodB*, and *fetAB* was confirmed by knockout and complementation. The result was shown as Supplementary Fig. 2 in the revised manuscript (page 7, line 140-142).

4. Per the presented model, one would expect $\Delta katG$ and or $\Delta katE$ mutants to be more susceptible to PG, but only the overexpression data is provided in figure 2a. Testing these strains could strengthen the argument.

Reply: Following your suggestion, we constructed the $\Delta katE$ and $\Delta katG$ mutants. However, they showed similar survival rate with the wild-type strain upon treatment with PG. Then, we constructed a double mutant of *katE* and *katG*, which is more susceptible to PG than wild-type *E. coli*. These results were added into the revised manuscript as Fig. 2f (page 9, line 175).

5. In the PG experiments where potentially toxic compounds are tested (for example Fe in 2d) or when important genes are deleted (like *rpoS* in 4b), it is essential to have a no-PG control for every condition. One could assume that deleting *rpoS* could affect cell growth and survival by itself. So, a $\Delta rpoS$ + PG must be compared to both WT + PG and $\Delta rpoS$ no PG. Regarding Fe, it is shown in 2f that Fe doesn't change the survival, however it's not mentioned what concentration of Fe is used here and no comparison can be made to 2g.

Reply: The experiment shown in Fig. 2a (previous Fig. 2d) was carried out

under no PG condition, and the result showed that overexpression of *fetAB* reduced intracellular iron levels in *E. coli*.

For Fig. 4b, survival of *E. coli* wild-type and *rpoS* mutant were tested without PG, and knockout of *rpoS* did not affect the growth of *E. coli*. This result was shown in Supplementary Fig. 4.

For clearer presentation, more information, including strains, overexpressed genes, and components used in the killing assay, were labeled in related figures in the revised manuscript.

6. It was clearly shown that ClpXP+*rssB* degrade RpoS. However, the role of RpoS in the process is still not clear. Throughout the manuscript, it was mentioned that RpoS is a major regulator of ROS response. While it does induce some ROS response genes, the major regulators during log phase (when these experiments are performed) are OxyR and SoxRS, and these two regulate the most important ROS response factors.

Reply: Our results showed that the RpoS protein level in *clpX* and *rssB* mutants was remarkably higher than in the wild-type *E. coli* strain, and the stabilized RpoS protein activates transcription of some ROS response genes. Based on your suggestion, we measured the mRNA level of *katE*, and found that it was 21 times higher in the *clpX* mutant than in the wild-type strain (Fig. 4d). Taken into account that overexpression of *katE* can protect *E. coli* from PG toxicity

(Fig. 2e), the role of RpoS in PG tolerance gets much clearer now. (page 13, line 255-257)

As SoxRS and OxyR are also major regulators regulating the defense system for oxidative stress in bacteria, their potential role in bacterial PG tolerance is discussed in the revised manuscript (page 23, line 461-463).

7. Relevant to point#6, recent work by Sen and Imlay (PMID 32601069) suggests that ClpXP degrades DPS and results in the release of iron. This fits beautifully to the data presented here: the $\Delta clpXP$ mutants have more intact DPS, and less free iron to react either PG and cause stress and death. Based on the presented data here and the published data, it seems more plausible that the effect of ClpXP in the process is through DPS and not (only) RpoS. I would highly recommend the authors to test this alternative hypothesis.

Reply: Thanks for your suggestion. To detect the protein level of Dps, a his₆-tag was fused to the c-terminal of Dps protein in its chromosomal locus. Under both conditions with the presence and absence of PG, the *clpX* mutant presented much higher level of Dps protein than the wild-type *E. coli* strain. This will help to reduce the intracellular concentration of free iron and the production of hydroxyl radicals. (page 13, line 258-263)

8. Lines 91 to 97 are repeats of lines 34 to 40.

Reply: The Abstract section has been rewritten in the revised manuscript (page 2, line 23-35).

9. The figures throughout the paper should be properly labeled. For example, fig 2a and 2b should have "WT" and " $\Delta sodB$ " labels. Also, all the assay components should be mentioned. For example, the addition of PG is mentioned in 2b and 2f, but not in the rest of the panels.

Reply: Thanks for the suggestion. More information, including strains, overexpressed genes, and components used in the killing assay, were labeled

in related figures in the revised manuscript.

10. A few typos and grammatical errors were present. Proofreading would be helpful.

Reply: Thanks for your suggestion. The manuscript has been carefully reviewed and revised throughout.

11. How much Fe is present in figure 2f?

Reply: In experiment shown in Fig. 2c (previous Fig. 2f), 1.15 mM Fe³⁺ was used. The concentration of Fe³⁺ was labeled in the revised figure.

12. What Fe containing compound is used?

Reply: In our previous study, 1.15 mM ferric ammonium citrate was used as iron supplier in PG production by the recombinant *E. coli* strain. So, 1.15 mM ferric ammonium citrate was still used in this study in *E. coli* adaptive evolution and killing assay. In susceptibility test of other species, 1.2 mM ferric chloride was used. This has been added in the revised manuscript (page 26, line 527).

13. Figure 4C: a control band of a housekeeping protein should be presented. Quantifying the data by image analysis would also be helpful.

Reply: A housekeeping protein, like GAPDH or actin, is usually used as a control band in eukaryotic cells, however no such protein is widely used in bacteria. Instead, SDS-PAGE of whole-cell lysate is usually used as loading control. Here, an SDS-PAGE was provided as Supplementary Fig. 5.

In Fig. 4c, relative intensity of each band was determined using the software ImageJ and shown in the revised manuscript.

14. Lines 558-9: RpoS is clearly visible in the WT strain, albeit at a lower amount.

Reply: It has been corrected accordingly (page 13, line 251).

15. KatE is probably the most important part of the RpoS regulon in response to ROS, and its mRNA amount should be presented in Figure 4d.

Reply: The mRNA level of *katE* was determined using quantitative RT-PCR, and it was 21 times higher in the *clpX* mutant than in the wild-type strain. This data was added in Fig. 4d.

16. The staining data in figure 7d needs some sort of quantification. It appears that the staining level is the same in PG and PG+Fe with the GPX4.

Reply: Thanks for your suggestion. The images were quantified using the ImageJ software with a plug-in tool IHC profiler, and the results were shown in

Supplementary Fig. 9.

17. No survival data is reported for figure 7.

Reply: In Fig. 7, all mice were sacrificed to weigh the tumors after 11-day treatment with PG, PG-iron complex and water, respectively. So, no survival data is showed in Fig. 7. This information was also added in the revised manuscript. (page 30, line 613)

REVIEWERS' COMMENTS:

Reviewer #1 (Remarks to the Author):

The authors Guang Zhao et al. submitted a revised version of their manuscript entitled "Phenolic compounds induce ferroptosis by promoting the hydroxyl radical generation in Fenton reaction" in order to be considered for publication in the journal "Communications Biology".

The authors acted on every concern and suggestion mentioned in the previous referee report and provided a point-by-point response.

For example, they have corrected the minor formal errors and inconsistencies that were criticized.

With regard to the suggestion to describe the quantification of the iron content in the medium, they refer to the production. An explicit analysis of the iron concentration is not and was not carried out.

Reference is made to the quantitative addition during the production of the medium. When referring to iron-free medium, the authors refer to the use of a minimal salt medium. That seems fine so far. In addition, the authors have repeatedly made improvements in the running text, which leads to even better clarity of the statements presented. This has eliminated inaccuracies. The revision of illustrations that were difficult to read has also improved the quality of the manuscript. In addition, the authors have cited more recent data/studies and also incorporated information on the induction of ferroptosis by iron complexes with phenolic structure into the current manuscript and embedded these findings in a meaningful way. This provides a more comprehensive presentation of previous and newly acquired scientific knowledge.

Overall, I congratulate the authors on this comprehensive study and encourage further processing of the manuscript. All the best!

Reviewer #2 (Remarks to the Author):

I am pleased to confirm that all the revisions and suggestions I made for the manuscript "Phenolic compounds induce ferroptosis by promoting the hydroxyl radical generation in Fenton reaction" have been thoroughly addressed and implemented. The authors have significantly improved the manuscript, enhancing clarity in descriptions, methodologies, and analyses.

The additional details provided in key areas, improved figure clarity, and the updated references notably increase the manuscript's strength. The paper now effectively communicates its significant findings and stands as a valuable contribution to its field.

Reviewer #3 (Remarks to the Author):

The revised manuscript has resolved almost all of the points that were raised in the original review. I would like to thank the authors for performing the experiments, and reinforcing their results.